# Long-range transport of air pollutants increases hazardous components of $PM_{2.5}$ in northern South America

Maria P. Velásquez-García[1,3,4], K. Santiago Hernández[2], James A. Vergara-Correa[1], Richard J. Pope[3,4], Miriam Gómez-Marín[1], and Angela M. Rendón[2]

[1]Grupo de Higiene y Gestión Ambiental , Politécnico Jaime Isaza Cadavid, Medellín, Colombia
[2]Grupo de Investigación en Ingeniería y Gestión Ambiental, Universidad de Antioquia, Medellín, Colombia
[3]School of Earth and Environment, University of Leeds, Leeds, UK
[4]National Centre for Earth Observation, University of Leeds, Leeds, UK

**Correspondence:** Maria P. Velásquez-García (eempvg@leeds.ac.uk)

**Abstract.** Long-range transport (LRT) of air pollutants, from a range of sources, can substantially enhance background pollution levels, especially in urbanized region, which can exacerbate high pollution episodes. In the Aburrá Valley (AV), Colombia, and other cities in Northern South America, biomass burning (BB), dust, and volcanic degassing have been identified as sources of LRT of aerosols. However, the impact of these sources on air quality and their characterization have yet to be thoroughly

studied. This work investigates the influence of these sources on the chemical composition of $PM_{2.5}$ during annual and intra-annual high-load aerosol events in the AV. We identified, tracked, and meteorologically characterized LRT events and evaluated their influence on $PM_{2.5}$ concentrations and its chemical composition. We found that the LRT of aerosols from BB, dust and volcanic degassing influenced approximately 13%, 8% and 13% of days through the year, respectively. We applied the Positive Matrix Factorization (PMF) statistical model for the different LRT event types (e.g. BB) to quantify the corresponding $PM_{2.5}$

concentration and chemical composition. For BB events, we identified high contributions from organic carbon (OC1, OC2), $F^-$ and secondary aerosol tracers, $SO_4^{2-}$ and $NO_3^-$. For dust LRT events, crustal mineral components, along with Ti and Ca, were the primary contributors to the aerosol composition, while $SO_4^{2-}$, Na, Al and Ca were the primary constituents during volcanic events. The concentration of some ions and toxic heavy metals (Cr, Mn, Cd, and Ni) were also elevated during BB and volcanic degassing events. BB exhibited the highest contribution of $PM_{2.5}$ within the LRT events ($\sim 11 \ \mu g/m^3$), while aerosols from

dust and volcanic events were also substantial ($< 7 \ \mu g/m^3$). Our study identifies the Orinoco and Middle Magdalena Valley as sizeable sources of BB aerosols and the Nevado del Ruíz volcano for volcanic aerosols. Additionally, we found that African dust approached the Andean region via the Caribbean route. As a result, we have identified the need for future chemical transport modeling studies in the region and new support strategies to manage internal and external pollution sources that degrade air quality in the AV and surrounding region.

## 1  Introduction

Long-range transport (LRT) of aerosols influences the chemical composition of air over thousands of kilometers (Kaneyasu et al., 2014; Wang et al., 2015; Rincón-Riveros et al., 2020) and plays a crucial role in the biogeochemical cycle of some

components, such as dust and biomass aerosols, which distribute iron and phosphorus across the oceans and continents (Okin et al., 2004; Boyd and Ellwood, 2010). Furthermore, aerosols interact with solar radiation, influencing cloud formation and light scattering (absorbance) with a cooling (warming) effect on the planet (Choobari et al., 2014). In particular, dust and black carbon can contribute to a reduction in the albedo over snow-covered regions, which can accelerate its melting, significantly affecting climate worldwide (Kaspari et al., 2014).

LRT of aerosols also increases human health risks, particularly in urban areas with high local emissions. Out of the different air pollutants, fine aerosols, represented by $PM_{2.5}$ (particulate matter with a diameter less than 2.5 $\mu m$), penetrate deep into the human body and trigger cerebrovascular and heart diseases, lung cancer and obstruction, and respiratory infections (Xie et al., 2021; Lippmann et al., 2013). Indeed, the LRT of aerosols can substantially increase the total $PM_{2.5}$ concentration to more hazardous levels and can increase the toxicity of the aerosols in respect to human health. For instance, carbonaceous dominant particles are known to be more toxic than crustal components (Tuomisto et al., 2008). Increased OC (organic carbon) and EC (elemental carbon) associated with biomass burning (BB) episodes have been linked to a greater risk of cardiovascular diseases (Hwang et al., 2017).

The accountability of $PM_{2.5}$ concentrations from LRT has urged cooperation between cities and countries to identify and control emissions of key precursor aerosol emission to reduce population vulnerability to hazardous air pollution events. For decades, in Europe and Asia, the LRT of air pollutants has been recognized as a significant factor of air pollution (Kulshrestha et al., 2014), and has informed the development and implementation of key policy strategies. For instance, the Task Force on Hemispheric Transport of Air Pollution (TF HTAP) is a collaboration led by the European Union and the United States that targets intercontinental and northern-hemisphere transport of particular matter (PM) and $O_3$, acknowledging the importance of these pollutants if air quality targets were to be met (UN, 2010). The collaboration has contributed to identifying regional sources, tracking, and basing more effective emission reduction strategies (Liang et al., 2018; Zhao et al., 2021; Dong et al., 2018). On a smaller scale, Hong Kong-Guangdong cooperation has addressed the transboundary issue of air pollution in the Pearl River Delta region (Zhong et al., 2013). This collaboration consisted of targeted of monitoring pollution levels and changes, evaluating the effectiveness of control measures, and providing feeback/training to stakeholders.

In Northern South America (NSA), the LRT of air pollutants is a more recently recognized public problem, and regional cooperation is gaining importance and momentum. Particularly in the Colombian Andes, biomass burning, desert dust, and volcanic emissions have been identified as three main sources of aerosols that can impact air quality from distant regions. Although for South America, biomass burning around the Amazon basin is one of the primary sources of aerosols (Ballesteros-González et al., 2020), atmospheric circulation patterns and substantial precipitation (i.e. aerosol wet deposition) limit the LTR of aerosols towards Colombia (Hamburger et al., 2013). Therefore, transboundary emissions from open fires in the Orinoco basin and the Caribbean are more important drivers of intra-annual periods of hazardous air quality for Colombian cities such as Bogotá, Medellín, Arauca, Yopal, Bucaramanga and Villavicencio (Mendez-Espinosa et al., 2019; Rincón-Riveros et al., 2020; Henao et al., 2021; Rodríguez-Gómez et al., 2022) and on Pico Espejo in Venezuela (Hamburger et al., 2013).

Likewise, the LRT of dust has also been reported in NSA, primarily emitted from the Sahara and Sahel deserts (Prospero et al., 2020). These particles predominately affect the Caribbean region, as reported in islands as Barbados, Guadalupe, Virginia,

and Tobago Island, and parts of the continent, e.g., French Guiana (Prospero et al., 2014; Kumar et al., 2014). African dust effects have also been reported in Colombia, with the analysis of high particle loads in June 2014 and 2020 (Bolaño-Ortiz et al.,

2023a; Mendez Espinosa et al., 2018; Bedoya et al., 2016). Notably, the LRT of African dust in June 2014 covered 95% of the country (dust concentrations $> 90 kg km^{-2}$) (Mendez Espinosa et al., 2018). However, a few long-term assessments have also been done in the Caribbean and Andean regions (Bolaño-Ortiz et al., 2023b; Arregocés et al., 2023).

Volcanic eruption and degassing also play a crucial role in the LRT of air pollutants in NSA and are part of the natural ecosystem in the region since it belongs to the Andean volcanic belt. Few studies have been conducted assessing the influence

of volcanic aerosols on air quality in the AV, but those which have mainly focused on the Sangay volcano in Ecuador and Nevado del Ruíz in Colombia (Casallas et al., 2024; Moran-Zuloaga et al., 2023). These studies identified elevated $PM_{2.5}$ and sulfur dioxide ($SO_2$) concentrations in the air from these volcanoes (Casallas et al., 2024; Cuesta-Mosquera et al., 2020; Trejos et al., 2021). The degassing activities from the Nevado del Ruíz has been especially highlighted for their magnitude and frequency (Carn et al., 2016).

In the last few years, multiple studies in Colombia have undertaken substantial efforts in monitoring and assessing the impacts of open fire burning emissions on local air quality (see e.g., Hernandez et al., 2019; Mendez-Espinosa et al., 2019; Ballesteros-González et al., 2020; Rincón-Riveros et al., 2020; Henao et al., 2021). Nonetheless, dust and volcanic aerosols studies have been limited to episodic high-pollution events (Mendez Espinosa et al., 2018) and do not provide a comprehensive long-term assessment of LRT of aerosols from these sources. (Liu et al., 2022). Overall, there are still significant gaps in our

understanding of how LRT of aerosols from these three sources influences regional air quality throughout the year. This relates directly to the absolute concentrations and the aerosol chemical composition. This is especially true for dust and volcanic aerosols, which are frequently overlooked due to their uncontrolled natural source type (Woo et al., 2020; Pouliot et al., 2012). Therefore, quantifying the properties (i.e. absolute concentration and chemical composition) of aerosols from LRT events would allow for a detailed assessment of the surface aerosol mixture (and air quality impacts) at local and national scales.

It would enable decision-makers to develop cooperative and effective projects to manage the risk of air pollution involving natural sources (Gómez Peláez et al., 2020; Jiao et al., 2021). Furthermore, advances in the characterization of the chemical composition of air can yield the opportunity to derive strategies to reduce the exposure of the population to high concentrations of certain species such as toxic heavy metals and carbonaceous matter (Gómez Peláez et al., 2020; Briffa et al., 2020; Allajbeu et al., 2017).

This study aims to analyze the impact of inter-annual LRT of biomass burning (BB-LRT), dust (Dust-LRT), and volcanic aerosols (Volcanic-LRT) on $PM_{2.5}$ concentrations and chemical composition in the Aburrá Valley (AV), which is one of the most populated metropolitan areas in Colombia, situated over the Andean Mountains. We characterize meteorologically favorable conditions for the LRT events and involve information from one of the largest registered $PM_{2.5}$ chemical characterization campaigns conducted in the region (April 2019 to October 2022), as well as in-situ $PM_{2.5}$ measurements and satellite-based

products. We identify representative events, analyze atmospheric transport, and highlight potential sources. Finally, we evaluate the impact of these sources on ground-level $PM_{2.5}$ concentrations and chemical composition in the AV.

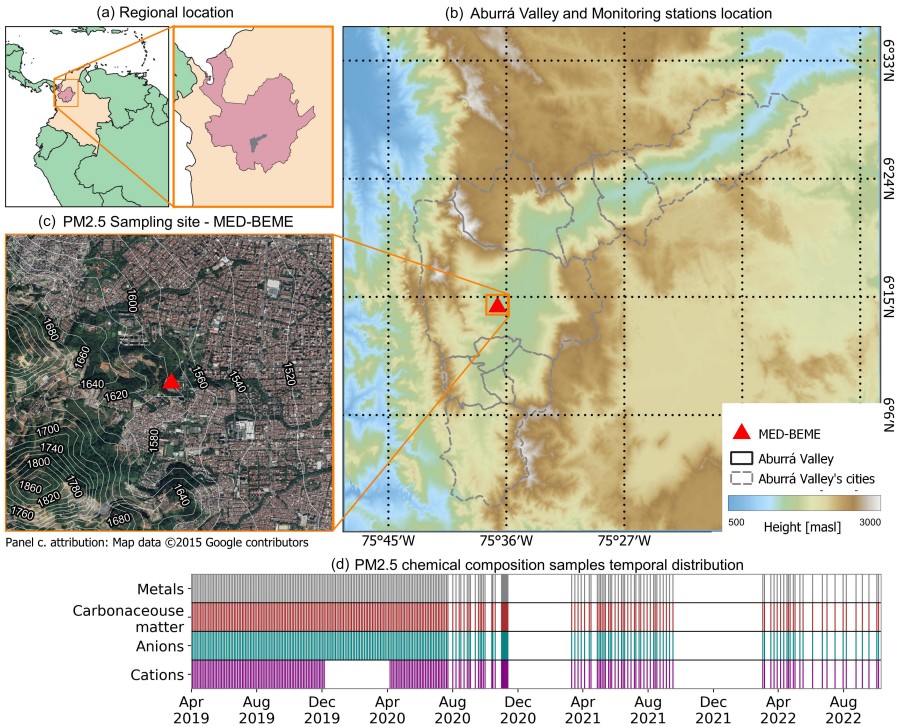

**Figure 1.** (a) Regional location of the AV, (b) Geomorphology of the AV and the distribution of the ten municipalities in black lines; (c) Location of the PM$_{2.5}$ chemical sampling campaign station (MED-BEME), represented by the red triangle on (b) and (c), and d. Temporal coverage of the PM$_{2.5}$ sampling campaign with white patches denoting no-sampling days.

## 2  Data and Methods

### 2.1  Study region

The AV, illustrated in Fig. 1b, is a 1.152 km$^2$ natural river basin located in the northeast of Colombia (see Fig. 1a). The territory contains ten cities, with Medellín as the largest city. The AV is in the central mountain range of the Colombian Andes, and its height ranges from 1300 masl (meters above sea level) in the valley to 2800 masl at the western mountaintop. By 2018, the population in the AV reached 3.73 million inhabitants, a dense conurbation. Due to the accelerated urban expansion (Echeverri and Orsini, 2011; Salazar Hernandez et al., 2022), the increasing vehicular fleet (Corrales Espinosa et al., 2016), and the limited air pollutant dispersion, the average daily concentrations of PM$_{2.5}$ frequently exceeds national and international standards (WHO, 2021, 15 $\mu g/m^3$) in the valley. Across the AV, more than 60% of days in the year exceeding these standards at most stations located in the urban areas (see Supplementary Table S1). Hence, the addition of external pollution sources caused severe air quality episodes in the valley (SIATA, 2021).

The AV has two rainy periods responding to the latitudinal migration of the Intertropical Convergence Zone (ITCZ), with maximum precipitation during April and November and minimum during January and July. The transition period, i.e., Febru-

ary and March, is characterized by persistent atmospheric stability and a thinner atmospheric boundary layer enhancing the accumulation of air pollutants (Herrera-Mejía and Hoyos, 2019). During this period of dry conditions, fires are more likely to spread in NSA, which affects air quality in the AV and neighboring regions (Mendez-Espinosa et al., 2019; Henao et al., 2021). These conditions modulate the intra-annual variability of $PM_{2.5}$ and $PM_{10}$ (Mendez-Espinosa et al., 2019). In this period, the transport and environmental local authority (Área Metropolitana del Valle de Aburrá) implementes special mobile sources control, limiting daily vehicle use.

## 2.2 Data sources

### 2.2.1 PM$_{2.5}$ chemical campaign

A sampling campaign was conducted at an urban background station from April 3, 2019, to October 5, 2022, to characterize the aerosol composition on $PM_{2.5}$ filters. A total of 247 daily (noon-to-noon) $PM_{2.5}$ samples were analyzed and recorded. While April 2019 to July 2020 represented an intense sampling campaign with samples every three days, the frequency of the surface site observations became less intense after July 23, 2020 (i.e. up to a maximum of 2 weeks during periods of routine sampling). However, there were two extended gaps in the campaign from November 2020 to mid-March 2021 and mid-September 2021 to March 2022. Despite the decrease in sampling frequency, the measurements still provide sufficient temporal coverage to get robust seasonal and annual information on aerosol mass concentration and composition.

The samples were taken in Belén, Medellín, on the western slope of the valley by the laboratory GYGHAM (Grupo de Higiene y Gestión Ambiental) from the Politécnico Jaime Isaza Cadavid. The sampling campaign was by MED-BEME, a station of the official air quality monitoring network (see Fig. 1c). The sector has a brick factory, but in general, the local economy is based on service and small businesses. The residential areas and brick factory are around 250 m and 620 m from the station, respectively. According to Gómez-Marín et al. (2021), the primary sources of $PM_{2.5}$ in MED-BEME are the ceramic industry (22.2%), emissions from BB (21.9%), diesel (13.6%), gasoline combustion (12.8%), incineration (1.7%), and coal-fired boilers (16.3%).

The campaign considered minerals (Be, Na, Mg, Al, Si, K, Ca, Ti, V, Cr, Mn, Fe, Co, Ni, Cu, Zn, As, Se, Mo, Ag, Cd, Sb, Ba, Hg, Pb), anions ($F^-$, $SO_4^{2-}$, $NO_3^-$, $Cl^-$), cations ($K^+$, $Mg^{2+}$, $Ca^{2+}$, $Na^+$), and carbonaceous matter species (OC1, OC2, OC3, OC4, OC5, Pyrogenic Carbon- PyC, EC1, EC2, EC3, EC4, EC5, EC6, OC, EC, C). In particular, cations were measured only in 207 of the 247 sampling days. Fig. 1d shows the temporal distribution of the samples throughout the study period. Note, that measurements of some minerals (e.g. Na, Mg, K, Se, Ca, and Hg) finished in March 2021. The elements were analyzed from an 8"x10" quartz filter in a High-volume $PM_{2.5}$ ambient air sampler (Reference: TE-6070, TISH), following the Australian/New Zealand Standard 3580.9.14:2013 Method 9.14 (AS/NZS, 2013). Distinct analytical methodologies were applied to determine the concentration of minerals, carbonaceous matter, and ions in the filters. An Inductively Coupled Plasma Mass Spectrometry (ICP-MS) methodology was used for minerals, a thermo-optical transmission (TOT) methodology for carbonaceous matter and an ionic chromatography (IC) for both anion and cations. $PM_{25}$ was additionally sampled by a Low-volume sampler (Reference: Wilbur TE-WILBUR - Tish) following the reference method described in the CFR 40 Appendix L

Part 50 suggested by the US-EPA (2011) and adopted by Colombian regulators (MinAmbiente-Colombia, 2010). The measured 24-hour PM$_{2.5}$ concentrations from the Low-volume are used for the Positive Matrix Factorization (PMF) statistical model. Further details on the techniques and the support of laboratory protocols are described in Gómez-Marín et al. (2021).

To complement the characterization of carbonaceous matter, the secondary organic carbon (SOC) was calculated using the elemental carbon trace methodology (Huntzicker et al., 1986). This method assumes that the organic carbon measured comes from the background, primary, and secondary sources and follows the relation described by Equations 1 and 2.

$$POC = OC_{back} + EC \times (OC/EC)_{pri} \tag{1}$$

$$SOC = OC_{Total} - POC \tag{2}$$

Where POC represents the primary organic carbon, OC$_{back}$ is the background OC, OC$_{total}$ is the consolidated OC measured in the campaign, and (OC/EC)$_{pri}$ is the primary ration between OC and EC.

Equation 1 describes a linear relation of primary carbon matter composition. The linear model was calculated using the 20$^{th}$ percentile OC and EC concentrations, aligned with the approach of Yao et al. (2020); Lin et al. (2009). The process was repeated for each season (DJF, MAM, JJA, and SON) due to the influence of weather conditions in the region. The resulting slope is interpreted as (OC/EC)$_{pri}$, and the intercept is the OC$_{back}$. After calculating (OC/EC)$_{pri}$ and OC$_{back}$, Equation 1 was used for calculating POC every sampling day using the measured EC. Then, the SOC was obtained with Equation 2.

This study involved creating four different linear regression models (Equation 1) for the two dry (June-August and December-February) and two rainy (March-May and September-November) periods. The square of correlation coefficient (R$^2$) showed high performance of the models with values $\geq 0.95$.

### 2.2.2 PM$_{2.5}$ concentrations

In addition to the campaign data, for the station MED-BEME, we downloaded hourly PM$_{2.5}$ concentrations from the official air quality monitoring network operated by the local early-warning system – SIATA (Sistema de Alerta Temprana de Medellín y el Valle de Aburrá, https://siata.gov.co/). From this, the PM$_{2.5}$ concentrations from April 2019 to October 2022 at the MED-BEME station coincided with the period and site of the chemical sampling campaign (illustrated in Fig. 1b). Daily average concentrations were calculated matching the sampling schedule (noot-to-noon). Only days with at least 75% of the hourly data were processed. Within our study period, we derived valid daily average concentrations for 96.7% of the days investigated.

We found a good agreement between the campaign (low-vol sampler) instrument and the official MED-BEME automatic station. For the study period, PM$_{2.5}$ concentrations from the automatic instrument had a minor overestimation against the reference method with a mean bias error of -0.76 $\mu g/m^3$. The corresponding mean absolute error (MAE) was 21.5%. For PM$_{2.5}$ measurements, the low-volume as a reference method provides better precision and accuracy than the MED-BEME sensor (Tasić et al., 2012), which follows an equivalent method. Despite this, the official sensor provides continuous measurements

that are used in this study for more robust comparisons. Regarding temporal variability, the Pearson correlation coefficient was 0.84, highlighting good consistency between them.

### 2.2.3 CAMS Reanalysis dataset

The European Centre for Medium-Range Weather Forecasts (ECMWF) provides atmospheric composition datasets through the Copernicus Atmospheric Monitoring Service (CAMS; Inness et al., 2019). For this research, we utilized the ECMWF Atmospheric Composition Reanalysis-4 (EAC4) dataset. It is a global atmospheric chemistry model, simulating a range of key tracers of composition, which uses a 4-dimensional variational data assimilation to assimilate satellite retrievals of, e.g., aerosol optical depth (AOD), carbon monoxide (CO), nitrogen dioxide ($NO_2$) and ozone ($O_3$) (Inness et al., 2019).

CAMS simulates different aerosols, which can be summed to determine the total AOD at 550 nm. To identify considerable loads from BB, dust, and volcanic aerosols, respectively, this study considers two AOD types: dust AOD (Du-AOD) and organic matter (OM-AOD), in addition to total column sulfur dioxide ($TCSO_2$). The products were downloaded daily at the original 3-hourly resolution and then resampled for local noon-to-noon periods aligned with the sampling campaign schedule. Only reanalysis information for the sampling location was used.

### 2.2.4 Meteorological data

Meteorological data come from the ECMWF Reanalysis v5 (ERA5; Hersbach et al., 2020). Fields of zonal and meridional winds, temperature, and moisture were obtained at temporal and spatial resolutions of 3 hours and 0.25° (approximately 27 km), respectively. ERA5 data were up-sampled to a daily frequency to guarantee consistency with the chemical sampling. Due to poor performance of reanalyses to simulate precipitation in the region (Posada-Marín et al., 2019), satellite-based fields were retrieved from the Global Precipitation Measurement (GPM) daily final precipitation product (Huffman et al., 2019) with a spatial resolution of 0.1°.

Further, back trajectories were exploited to estimate pollutants arriving at the chemical sampling point in the AV (see Fig. 1c). 8-day trajectories starting at heights of 800 hPa, 750 hPa, and 700 hPa were run every 3 hours with a time step of 3 hours. The basic model was created with SIATA and has been used in other investigations like Hoyos et al. (2019) and Pérez-Carrasquilla et al. (2023). This model only uses wind data from the product's 4h-daily pressure levels, U-wind, V-wind, and omega (with a spatial resolution of approximately $2.5^o$) from NCEP-NCAR Reanalysis 1 (https://psl.noaa.gov). The model follows Equation 3. for estimating every back position.

$$X(t - \Delta t) = X(t) - V(X, t)\Delta t \tag{3}$$

Where X is the location with vertical and horizontal coordinators, V is the wind vector, and t is the time.

## 2.3 Identification and tracking of aerosol events

### 2.3.1 Identification of LRT events

Despite the selected CAMS products covering the whole atmospheric columns and not only the surface, CAMS has a high
predictive capability for $PM_{2.5}$ in the AV (Pérez-Carrasquilla et al., 2023). Furthermore, the CAMS' products reasonably
capture $PM_{2.5}$ tendencies and extreme events in the territory (Casallas et al., 2022). Therefore, the magnitude of CAMS
products OM-AOD, Du-AOD, and $TCSO_2$ in the AV were utilized to identify possible BB-LRT, Dust-LRT, and Volcanic-LRT
events, respectively.

To help identify $PM_{2.5}$ days that were subject to LRT events, the time series of the CAMS data sets were standardized.
Here, for each day, the time-series average (mean) was subtracted from the daily value and then normalized by the time-series
standard deviation (i.e., $1\ \sigma$). This then allowed for the identification of LRT events using a range of subjective thresholds
and the quantification of the corresponding $PM_{2.5}$ concentrations and composition. The thresholds investigated ranged from
$0.8$ to $2.0\ \sigma$, but in this study, we present the results for thresholds of $0.8$, $1.5$, and $2.0\ \sigma$. A 7-day rolling window was used
to accurately identify prolonged and intense periods of LRT events. Within this window, at least 4 days had to have values
above the respective thresholds to be classified as a LRT event. We subjectively chose 4-days of elevated values due to the
sampling frequency of the campaign. Here, campaign temporal sampling was $\geq 3$ days, so these criteria were required to get
representative samples of the aerosol composition for the chemical characterization of the sources.

After identifying potential LRT events, the center day of the 7-day window was marked as the peak of the LRT event. Then,
every day was labelled according to their distance to the peak of the LRT event as n days to the event ($DtE_n$). For this, the peak
of the event is labled as $DtE_0$ and the LRT events ranged from 3-days before ($DtE_{-3}$) to 3-days after ($DtE_3$) the peak event.
The days outside of the 3-day windows around a $DtE_0$ are typically identified as no-event days and assumed to be independent
of LRT events. Note that this methodology allows continuous days to be marked as peaks for longer-lasting events.

Although the process described above was run for every event and threshold, the results in this study are based on $0.8\ \sigma$ for
BB-LRT and $1.5\ \sigma$ for Dust-LRT and Volcanic-LRT, as is described in Section 3.1.

### 2.3.2 Regional meteorological analysis

A regional analysis of meteorological fields was performed during the LRT events to identify atmospheric conditions that
favored aerosol transport from the sources of interest. Here, we derived the anomalies for multiple meteorological variables
by taking the values of each pixel and subtracting the average value of the corresponding month. Then, composites during
days with different LRT events (i.e. days in the range of $DtE_{-3}$ to $DtE_3$) were calculated for both meteorological fields and
anomalies. This approach helped to reveal particular meteorological conditions during days with aerosol LRT events. The
results are also supported by analyzing air masses arriving at the AV using back-trajectories during the LRT events. The
number of times trajectories pass through a grid box was counted and the percentage of occurrences calculated yielding an
estimate of the probability that a particular grid box will experience a LRT event for each type.

### 2.3.3 Influence on local PM$_{2.5}$ concentrations

The local measurements of PM$_{2.5}$ were used to assess the influence of CAMS-based identified LRT events on air quality in the AV. The daily PM$_{2.5}$ datasets were labeled with the days to the closest event peak (DtE) to compare concentrations before and after the events. The days before the event constitute from 15 to 4 days before the event peak (DtE$_{-15}$ to DtE$_{-4}$), while the days after the event peak consisted of 4 to 15 days after the event peak (DtE$_4$ to DtE$_{15}$). These ranges were determined to ensure enough data from the chemical campaign from similar weather conditions for comparison between the days affected

by the LRT of aerosols and the days before and after. We used the Mann-Whitney U test to compare these periods. The null hypothesis is denied with the confidence of 90% (p-value $\leq$ 0.1) and 95% (p-value $\leq$ 0.05).

The COVID-19 lockdown period (April 1st to June 1st, 2020) was filtered from our record to avoid perturbation out of standard emission patterns in the region.

### 2.3.4 PMF and PM$_{2.5}$ chemical composition

The PMF is a receptor model developed by the EPA (Environmental Protection Agency) to analyze water and air samples. The model expresses observations of PM$_{2.5}$ as a sum of the contributions from a number of source profiles (non-time dependent). In this case, we are using the PMF model and our campaign measurement data of PM$_{2.5}$ chemical composition and PM$_{2.5}$ concentrations to identify the dominant sources (e.g., coal combustion, vehicular emissions, secondary pollution) during each LRT event (i.e., BB, dust or volcanic LRT events). For the AV, 5 to 7 factors (i.e., sources) have been suggested (see Gómez-

Marín et al. (2021)).

The PMF is mathematically expressed as (Paatero and Tapper, 1994):

$$X_{ij} = \sum_{k=1}^{N} g_{ik} f_{kj} + e_{ij} \tag{4}$$

The objective of PMF is to determine the values of g$_{ik}$, f$_{kj}$, and $N$ (i.e. the number of factors) that best reproduce $X_{ij}$. Here, $X$ represents the measurement data for sample $i$ (daily temporal resolution) and chemical component $j$. An inversion

approach is used for this matrix problem, exploiting an iterative scheme to converge on the solutions for $g_{ik}$ and f$_{kj}$. In the factorization problem, $f_{kj}$ contains the concentration of each chemical component ($j$) in the unit profile for the factor $k$ (i.e. PM$_{2.5}$ source). The matrix $g_{i,k}$, the contribution factor, defines how much of the profile is counted in the total concentration in day $i$. $e_{i,j}$ contains the residual for each compound/sampling day. Outputs from the PMF include statistical metrics that help to evaluate the model performance, such as the correlation coefficient (R$^2$), the recuperated mass (% RM), and the objective

function value (Q). The PMF adjusts $g_{ik}$ and $f_{kj}$ to minimize the function Q in Equation 5 (Paatero, 1997).

$$Q = \sum_{i=1}^{n} \sum_{j=1}^{m} \left[ \frac{X_{ij} - \sum_{k=1}^{N} g_{ik} f_{kj}}{\mu_{ij}} \right]^2 \tag{5}$$

Where $\mu_{ij}$ contains the uncertainty compound/sampling day.

The model uses two input uncertainties. The first is the species uncertainty ($\mu_{ij}$), which does not change in the process and is according to the unique method used for every species sampled. In this study, $\mu_{ij}$ considers the measurement and analytical errors, comprising uncertainties of the volume of air processed, filter area, and species mass, in addition to the uncertainty of the detection limit of the measurement method. The last one represents the analytical uncertainty and was calculated following Noris and Duvall (2014) by using the corresponding method's detection limit suggested by Eugene Kim and Qin (2005). However, for PM$_{2.5}$, we followed the approach of Eugene Kim and Edgerton (2003) who set $\mu_{ij}$ (where $j$ represents PM$_{2.5}$ for day $i$) greater than the daily PM$_{2.5}$ concentration (i.e., three times the concentration in this study). The second uncertainty in $\mu_{ij}$ is recognized as the "Extra modeling uncertainty". It is added to the model as a percentage to cover other inherent errors, such as variations of source profiles and chemical transformations in the atmosphere. A range from 10% to 16% was tested for PMF for each type of LRT event, which aligns with the approach of other studies (Callén et al., 2009; Shin et al., 2022; Salim et al., 2019).

During the modeling process, species can be classified as "strong", "weak," or "poor" based on individual statistics. The "weak" category triples the uncertainty $\mu_{ij}$, and the "bad" category excludes the species from the model. Two statistics are key for categorizing the species. First, the signal-to-noise ratio (S/N) represents overall uncertainty per concentration. S/N<0.5 is generally classified as "bad" because the uncertainty surpasses two times the concentration (Noris and Duvall, 2014). Second, the regression diagnostic where R$^2$ provides information about the linear relationship between the observation and modeled concentrations for each species.

The model convergence evaluation statistics considers this classification with the Q/Q$_{expected}$ ratio. Q$_{expected}$ is the theoretical value of Q (Equation 5), expressed as Equation 6.

$$Q_{expected} = (n \times m_s) - ((N \times n) + (m_s \times N)) \tag{6}$$

where $n$ is the number of sampling days, and $m_s$ is the number of "strong" species used for modeling (Noris and Duvall, 2014).

We expected relatively small temporal samples (i.e. number of days) of the measured pollutants for each LRT type in the PMF model since the targeted LRT events would not cover the majority of the days during the PM$_{2.5}$ chemical campaign. Nonetheless, multiple studies with PMF samples ranging from 14 to 30 have reported useful and meaningful results (Yu et al., 2015; Haghnazar et al., 2022; Via et al., 2022). As the sample dataset decreases, rotational ambiguity caused by infinite valid solutions strongly affects the results and increases overall uncertainty (Manousakas et al., 2017). To mitigate the error, the software EPA PMF v 5.0 allows for estimating the effect of random errors and rotational ambiguity in the dataset using bootstrapping (BS) and Displacement (DISP) tools. While BS evaluates random errors by performing 100 runs with randomly relocated blocks of observation of the original dataset, DISP focuses on indicating rotational ambiguity by adjusting up and down all values in the factors profile. This is restricted to 4 permitted changes when calculating Q (dQmax) and monitoring major factor swaps (Noris and Duvall, 2014).

**Table 1.** Statistics of OM-AOD, Du-AOD, and TCSO$_2$ representing BB-LRT, Dust-LRT, and Volcanic-LRT

| Threshold | Events stats. | OM-AOD | Du-AOD | TCSO$_2$ |
|-----------|---------------|--------|--------|----------|
| 0.8 $\sigma$ | Samples | 31 | 31 | 59 |
| 1.5 $\sigma$ | Samples | 18 | 19 | 32 |
| 2.0 $\sigma$ | Samples | 10 | 4 | 21 |
| 0.8 $\sigma$ | Limit | 0.22 | 0.01 | $0.70 mg/m^2$ |
| 1.5 $\sigma$ | Limit | 0.28 | 0.02 | $0.87 mg/m^2$ |
| 2.0 $\sigma$ | Limit | 0.32 | 0.02 | $0.99 mg/m^2$ |
| 0.8 $\sigma$ | Average | 0.25 | 0.02 | $0.78 mg/m^2$ |
| 1.5 $\sigma$ | Average | 0.29 | 0.02 | $0.88 mg/m^2$ |
| 2.0 $\sigma$ | Average | 0.32 | 0.03 | $0.97 mg/m^2$ |

"Samples" represent the number of sampling days available for the PMF characterization.
Each threshold limit defines the calculated absolute magnitude. Note, AOD is a
dimensionless quantity.

Additionally, constraining the base run can improve the solution when data is limited by reducing the rotational space (Dai et al., 2020). The PMF software has the functions to "pull down maximally," "pull up maximally," "set to zero," and manually set the profile concentrations. While the first two options are soft constraints, the third and fourth are hard constraints and require a high level of confidence in the magnitude of the profile contributions. For this study, only soft contains are contemplated. The constraint increases the final Q value, which should be less than 5% (i.e., %dQ < 5%), the recommended maximum change (Noris and Duvall, 2014). For this study, the %dQ was set by default to <0.5%.

Due to missing data, all species except cations are included in the model. Only total OC and EC from carbonaceous matter are included for Duts-LRT and Volcanic-LRT events since comprising all carbon species could particularly weigh those and overshadow tracer elements for the sources. For those LRT events, the potential run period is reduced until April 2021 since the tracer minerals Na, K, and Mg were not measured afterwards.

The factor interpretation of this study was primarily supported by the source characterization made by Gómez-Marín et al. (2021) for the studied station.

A final comparison of the characterized compounds was conducted, focusing on the days when a positive contribution occurred for the LRT events ($g_{i,k}$>0) and the days categorized as before (DtE$_{-15}$ to DtE$_{-4}$) and after events (DtE$_4$ to DtE$_{15}$).

## 3 Results

This section is divided into four subsections. The first subsection identifies potential aerosol events from BB-LRT, Dust-LRT, and Volcanic-LRT. The second is focused on the temporal and regional characterization of these events, including a description of the associated meteorological patterns and possible aerosols sources. Finally, the third and fourth subsections locally assess the identified events using ground-level PM$_{2.5}$ concentrations and chemical compositions, respectively.

## 3.1 Identification of LRT events

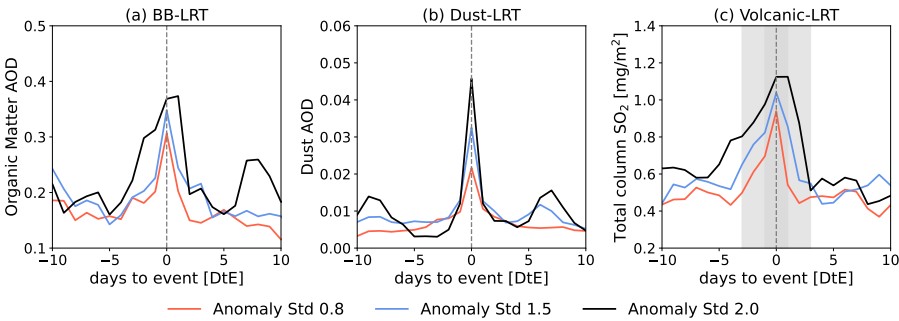

**Figure 2.** Average magnitudes of OM-AOD (a), Du-AOD (b), and TCSO$_2$ (c) concentration around the events. Different colours represent different anomalies. The shadowed region delimits the event. Du-AOD is on a scale that is ten times lower than OM-AOD.

The products OM-AOD, Du-AOD, and TCSO$_2$ from the CAMS reanalyses were used to identify aerosol from BB-LRT, Dust-LRT, and Volcanic-LRT, respectively. Table 1 shows different statistics about the identification for each type of event using the threshold of 0.8 $\sigma$, 1.5 $\sigma$, and 2.0 $\sigma$. The most intense events reached maximum magnitudes of 0.66 for OM-AOD (March 30, 2020), 0.11 for Du-AOD (June 24, 2020), and 1.96 $mg/m^2$ for TCSO$_2$ (September 10, 2022). On these days, the products exceeded the average magnitudes for the target events more than twice for OM-AOD and TCSO$_2$ and more than five times for Du-AOD. Notably, OM-AOD reached 77% of the total AOD in the most intense event. Regarding the variability, all identified events showed high variability with a standard deviation around 42%, 90%, and 35% of the average for OM-AOD, Du-AOD, and TCSO$_2$, respectively.

Figure 2 shows the average behavior of each variable centered around the event peak for the threshold of 0.8 $\sigma$, 1.5 $\sigma$, and 2.0 $\sigma$. The events exhibit a clear peak centered on DtE$_0$ for most of the considered thresholds. However, events identified with 2.0 $\sigma$ exhibit noiser behaviour around the peak for BB-LT and volcanic-LRT. In particular, OM-AOD exhibits a second peak after the event peak. This could be caused by either the smaller sampling size or higher aerosol loads that do not meet the duration criteria for the threshold. Besides, the strictest thresholds (2.0 $\sigma$) result in a considerable reduction of studied events (see Table 1), which is critical for the assessment of the $PM_{2.5}$ chemical composition.

For each type of LRT transport event, we selected a single $\sigma$ threshold value. Based on the results and literature review about the expected frequency of each type of event for the city (Mendez-Espinosa et al., 2019; SIATA, 2021), we selected events that exceed 0.8 $\sigma$ for BB-LRT and 1.5 $\sigma$ for Dust-LRT and Volcanic-LRT. Anomalies over 2.0 $\sigma$ significantly limited the number of days for the analysis, whereas 0.8 $\sigma$ probably over sampled volcanic and dust aerosol load events since these are recorded as extremely infrequent events. Moreover, the selection ensures annual represenation for each event (see supplementary Fig. S2). BB-LRT, Dust-LRT, and Volcanic-LRT events were all detected during the complete study period, occurring in at least 3-years of the record each. These events typically occurred for 13%, 8%, and 13% of days in the year, respectively. With this selection, the sampling campaign can represent BB-LRT, Dust-LRT, and Volcanic-LRT events in the PMF with 31, 19, and 32 samples.

## 3.2 Regional analysis of events

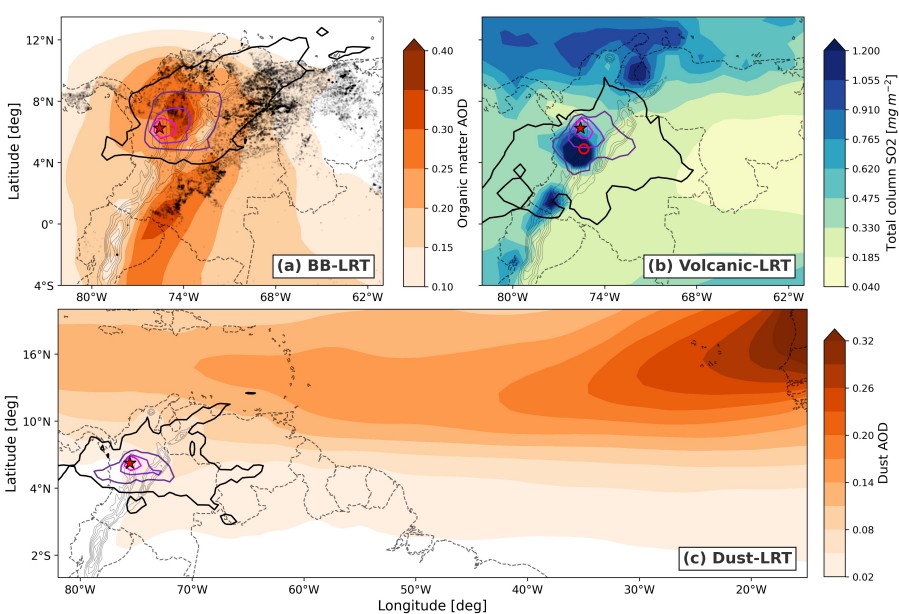

**Figure 3.** Mean spatial distribution of OM-AOD (a), TCSO$_2$ (b), and Du-AOD (c) for days with each type of LRT event. Pink-to-black contours enclose the regions from which 20, 10, 5, and 1% of air masses arrive at the AV according to the back-trajectories. Gray dots in a. show MODIS-retrieved hotspots associated with fires with a 70% of confidence. The red star marks the location of the AV, while the red circle in the upper right panel is the location of the Nevado del Ruíz Volcano. The black contours show the terrain elevation from 1500 to 5000 masl.

Regional analysis was conducted to establish the meteorological conditions that may favour air pollution LRT events and identify potential aerosol sources. The average spatial distribution of the identified AOD-OM, Du-AOD, and the TCSO$_2$ during the aerosol LRT events is shown in Fig. 3, with the integrated back trajectories of air masses arriving at the AV. OM-AOD values above 0.2 occur over most of Colombia during BB aerosol events (Fig. 3 a), consistent with a high number of fire hotspots in northern South America (Venezuela and Colombia). In particular, the back trajectories during these days suggest that a large part of the air masses arriving at the AV (10 to 20 %) come from the northeast, where OM-AOD exhibits its peak values (exceeding magnitudes of 0.35).

Volcanic-LRT (Fig. 3 b) shows high SO$_2$ values (over $1.2\ mg\,m^{-2}$) in the atmospheric column above the Nevado del Ruiz volcano (red hollow circle). Additionally, another two spots are evident: the first one, towards southwestern Colombia, could be related to another degassing volcano called the Galeras, whereas the other one is on Maracaibo Lake, with relatively high values are possibly associated with oil extraction (see Fioletov et al. (2016)). According to the back-trajectories, during Volcanic-LRT events, around 10% of the air masses arriving in the AV come from the southeast, where the Nevado del Ruiz hotspot exists. A smaller fraction of air masses (1%) arrive from the hotspot (i.e., Galeras volcano) in the southwest of Colombia. Similarly,

there is clear propagation of dust aerosol from the Sahara desert towards the Caribbean and, to a lesser extent, NSA (see Fig. 3 c). While air masses for BB-LRT and Volcanic-LRT have different directions, most air masses that reach the AV during Dust-LRT events come from the east.

Despite the short study period, the identified events suggest a marked annual cycle (Fig. 4). During March, 50.0% of studied days represent the BB-LRT. April and February also show high proportions BB-LRT events, with 43.3% and 24.8%, respectively. Concerning the dust events, two seasonal peaks are observed, April and July, with a proportion of 23.3% and 23.4%. The distribution for volcanic-LRT ranges from June to September, with a clear peak in August with a frequency of 49.2%. Additionally, as shown in Fig. 4, external events overlap in some cases. The largest occurrence of combined LRT events was observed in April comprising BB-LRT and Dust-LRT events, while BB-LRT and volcanic-LRT events showed substantial overalp in September. Here, where days were defined as both, e.g., BB-LRT and Dust-LRT, they were added to the joint classification (i.e., BB-LRT & Dust-LRT) and were not included in the respective singular classifications. Overall, the annual cycle of these LRT event occurrences and the local PM$_{2.5}$ concentration shows that the influence of LRT on local on measured PM$_{2.5}$ is non-linear.

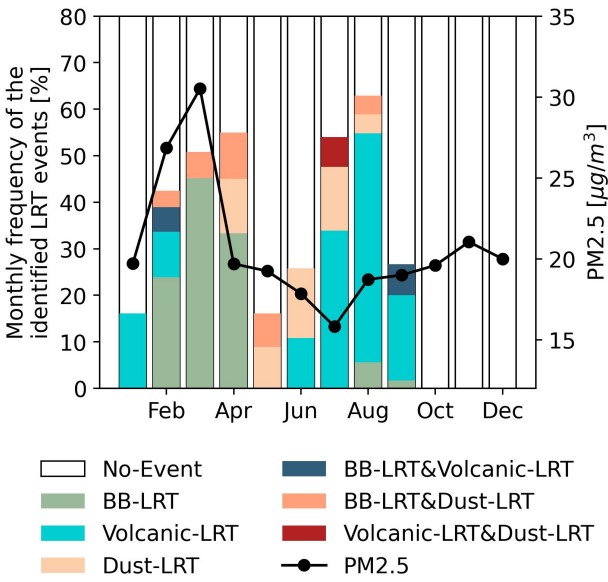

**Figure 4.** Monthly frequency of days with BB-LRT (green bars), Dust-LRT (wheat bars), and Volcanic-LRT (blue bars) events as identified from the CAMS reanalysis. Overlapped events are depicted in dark blue (BB and Volcanic), orange (BB and Dust), and red (Volcanic and Dust) bars since different LRT events can happen at the same time. White bars represent the frequency of days without LRT events, while the black line shows the monthly average PM$_{2.5}$ concentration ($\mu g/m^3$) for the MED-BEME station.

Although LRT events display a marked seasonality, a significant percentage of days in each month experience negligible impacts from LRT events (see Fig. 4). This suggests that intra-seasonal variations are also relevant in explaining the occurrence of these events. To isolate specific meteorological patterns during LRT events days, Figs. 5 and 6 present composites of

meteorological variables for the low (800 to 700 hPa) and mid (600 to 400 hPa) troposphere for the different kinds of events. The Andes mountain range generally causes a large spatial variability in the low-level wind field. The winds from the east of the Andes cross the mountain range through two zones with relatively low altitudes (hereafter mountain passes), which are demarcated with white boxes (see Figs. 5 a, b, c).

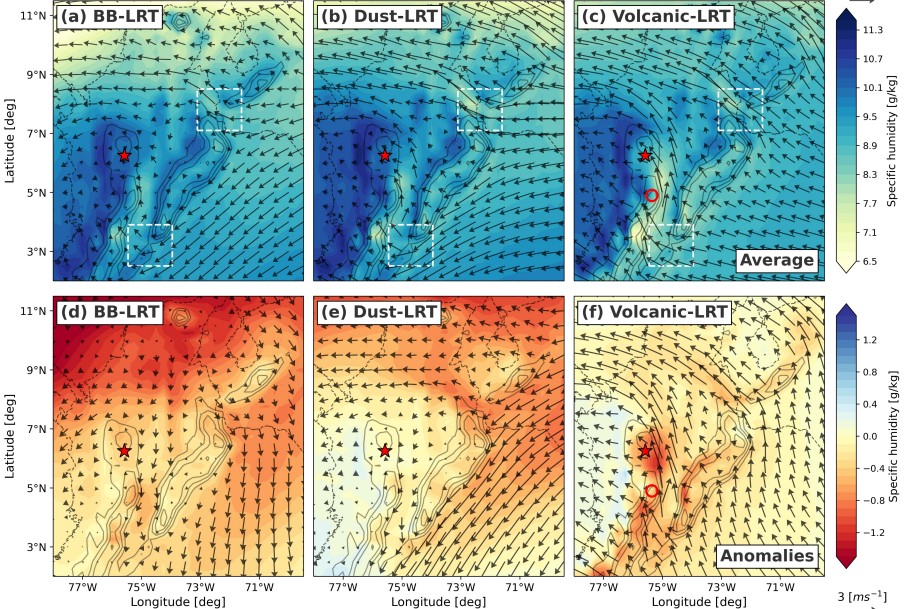

**Figure 5.** Meteorological composites for low-level (800 to 700 hPa) winds and specific humidity during days with (a) BB-LRT, (b) Dust-LRT, and (c) volcanic-LRT events. d, e, f: as upper panels but for anomalies. A red star marks the Aburrá Valley, and the red circle in the upper right panel is the Nevado del Ruíz volcano. The white rectangles are two mountain passes in the Colombian Andes. The black contours show the terrain elevation from 1500 to 5000 masl.

Anomalous dry winds of around 1 m/s originate from the north during BB-LRT events (Fig. 5 d), connecting the AV with the high OM-AOD region. Additionally, winds blowing from Venezuela, where multiple BB hotspots were identified, reach northern Colombia through the mountain pass located north of the Andes, at the border between Colombia and Venezuela (see Fig. 5 a). Low-level winds during LRT events are in agreement with the back trajectories of Fig. 3 a. A reduction of 2 to 3 m/s in the intensity of mid-level winds (between 5°N and 8°N; see Fig. 6 a, d) was also observed along with less rainy conditions (anomalies of around -2 mm/d) in northern Colombia.

For Dust-LRT events, low-level winds flow across the two mountain passes in the Andes (Fig. 5 b) with increased easterly winds north of Colombia and northeasterly winds east of the Andes (Fig. 5 e), representing a dry flow from the Atlantic Ocean where Du-AOD is higher. Figs. 6 b and e, show that mid-level winds have notable eastward direction with anomalous winds coming from the Caribbean and a rainfall reduction between 3 to 4 mm/d in NSA.

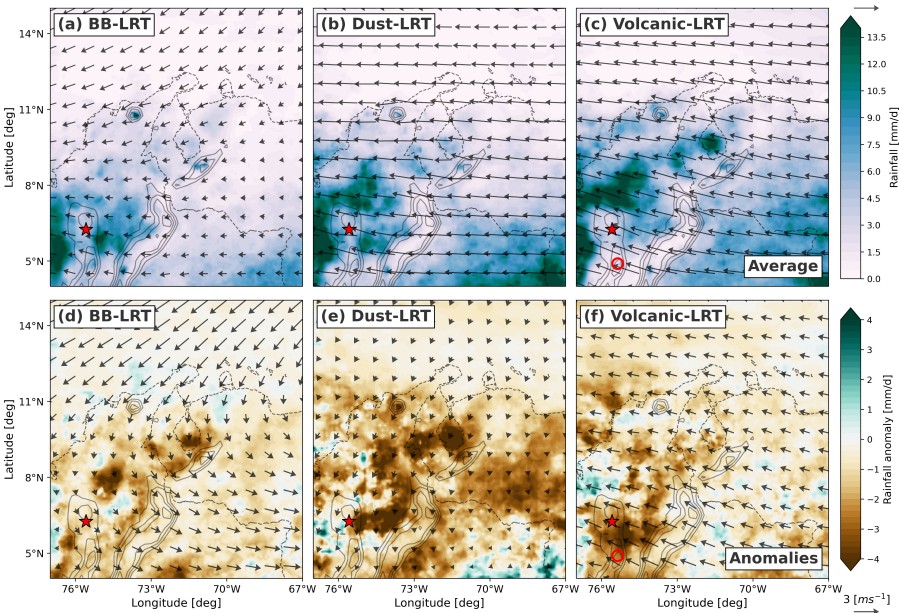

**Figure 6.** Meteorological composites for mid-level winds (600 to 400 hPa) and rainfall during days with (a) BB-LRT, (b) Dust-LRT, and (c) Volcanic-LRT events. d, e, f: as upper panels but for anomalies. The red star marks the Aburrá Valley, and the red circle in the upper right panel is the Nevado del Ruíz volcano. The black contours show the terrain elevation from 1500 to 5000 masl.

Finally, days with volcanic-LRT events are characterized by southeasterly winds traveling toward the AV (see Fig. 5 c) from the Nevado del Ruíz volcano region. Here, the southeasterly wind anomalies are 3 m/s yielding stronger transport across the southern mountain pass, as shown in Fig. 5 f, along with drier air reaching the AV. Less rainy conditions are also present in the Colombian Andes (Fig. 6 f), particularly between the AV and the Nevado del Ruíz volcano where a reduction in rainfall of around 3 mm/d is observed.

Overall, the three types of events are accompanied by less rainy and drier atmospheric conditions in northern Colombia, Venezuela, and the Andes, respectively. Here, rainfall and specific humidity anomalies are typically larger (in absolute terms) than -4 mm/d and -1 gkg$^{-1}$, respectively. These results highlight the importance of precipitation control on aerosol LRT to the AV (i.e., lack of precipitation aids the LRT of aerosols).

### 3.3 PM$_{2.5}$ concentrations change

After identifying LRT events and the meteorological conditions favouring them, an analysis of local surface PM$_{2.5}$ was undertaken to understand the influence of LRT on the AV's air quality. Figure 7 compares the PM$_{2.5}$ concentrations before (DtE$_{-15}$ to DtE$_{-4}$), during (DtE$_{-3}$ to DtE$_3$) and after (DtE$_4$ to DtE$_{15}$) each event. BB-LRT had the highest PM$_{2.5}$ increments, especially at DtE$_0$. 86.7% of values during the peak day exceed the average PM$_{2.5}$ for the study period (20.25 $\mu g/m^3$). Over half of PM$_{2.5}$ concentration during DtE$_0$ exceeds twice the WHO (2021) guidelines for daily concentrations (15 $\mu g/m^3$). At

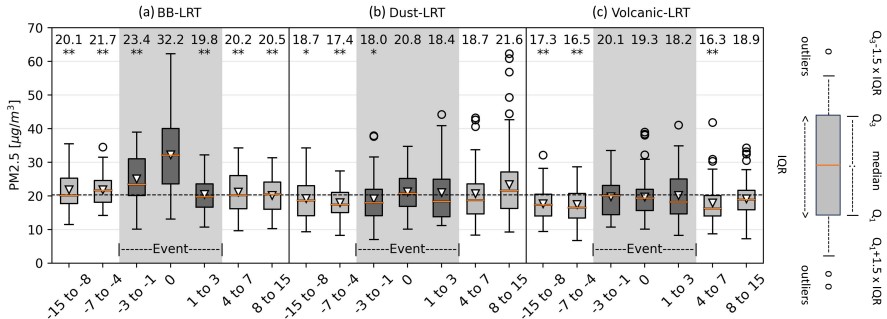

**Figure 7.** PM$_{2.5}$ concentrations before, during and after (a) BB, (b) dust and (c) volcanic aerosols events. The chart's top presents the median concentration of each dataset, along with the significance level compared with the peak of the data, $\leq 0.1$ (*) or $\leq 0.05$ (**). The shadowed darker boxes represent the events. The down triangles represent the average concentration for each day range, and the grey dashed line represents the total study period average. The boxes are limited by the first and third quartile, Q$_1$ and Q$_2$

DtE$_0$, the PM$_{2.5}$ average concentrations (31.48 $\mu g/m^3$) were significantly higher than the surrounding days (p-value $\leq 0.05$), according to the Mann-Whitney U test. Similarly, the event days before the peak (DtE$_{-3}$ to DtE$_{-1}$) had significantly higher concentration compared to days before and after the identified BB-LRT events (p-value $\leq 0.05$). In contrast, concentrations following the event peak are similar to the average magnitude, indicating a sudden decrease.

While there is a substantial BB-LRT signature in the local PM$_{2.5}$ measurements, the response is more subtle for Dust-LRT and Volcanic-LRT events. For Dust-LRT and Volcanic-LRT events, the median PM$_{2.5}$ concentrations during the event (Dte$_{-3}$ to Dte$_3$) are actually lower than the campaign average (20.25 $\mu g/m^3$). However, for Dust-LRT events, the PM$_{2.5}$ concentrations are significantly (p-value $\leq 0.1$) higher than before the event. This is likely due to the large variability in the Dust-LRT event data with the Q$_3$ quartile peaking over 35 $\mu g/m^3$, which was lower before the event (especially Dte$_{-7}$ to Dte$_{-4}$ reaching

approximately 28 $\mu g/m^3$). For Volcanic-LRT events, there is a small PM$_{2.5}$ gradient between days before and during the LRT event, which is not dissimilar to that of the Dust-LRT events. Although, the tighter range in the variability (i.e. lower Q$_3$ values) yields a non-significant difference despite the larger Volcanic-LRT event median values (e.g. by approximately 2-3 $\mu g/m^3$). After the peak Dust-LRT event, PM$_{2.5}$ in the Dte$_4$ to Dte$_7$ window decreases to pre-event levels. However, the Dte$_8$ to Dte$_{15}$ shows a similar median value to the Dust-LRT peak (DtE$_0$) at approximately 20 $\mu g/m^3$ and has a larger data range

(i.e. Q$_3$ peaks at approximately 43 $\mu g/m^3$). These larger than expected values occurred in March 2020, which was a period strongly affected by BB-LRT events. For days (Dte$_4$ to Dte$_7$) after Volcanic-LRT events, we find that PM$_{2.5}$ concentrations are significantly lower (p-value $\leq 0.1$), contrasting with the subsequent days (Dte$_8$ to Dte$_{15}$). Overall, it is clear that the BB-LRT events have a larger influence on PM$_{2.5}$ concentrations in the AV than Dust-LRT and Volcanic-LRT.

## 3.4   PMF and chemical composition change

The PMF models of the BB-LRT and Volcanic-LRT presented a good fit, and all residuals are normally distributed; however, for the Dust-LRT, some components' residuals exceed the recommended range of -3 to 3 (Noris and Duvall, 2014), so 3 samples

**Table 2.** Statistics performance of PMF for models during days affected by BB-LRT, Dust-LRT, and Volcanic-LRT

| Properties | BB-LRT | Dust-LRT | Volcanic-LRT |
|---|---|---|---|
| Model statistics | | | |
| Samples | 31 | 16 | 32 |
| N. factors | 6 | 6 | 6 |
| Non-weak species | 32 | 27 | 27 |
| Bad species | OC5,EC6 | Be | — |
| $Q/Q_{expected}$ | 0.99 | 1.00 | 0.93 |
| Add. uncertainty[%] | 16 | 11 | 12 |
| BS mapped factor [%] | 82 | 82 | 88 |
| DISP swaps | 0 | 0 | 0 |
| Model statistics for $PM_{2.5}$ | | | |
| $PM_{2.5}$ error [$\mu g/m^3$] | 2.50 | 1.84 | 2.60 |
| $R^2$ | 0.92 | 0.90 | 0.73 |
| RM [%] | 99.31 | 98.24 | 97.50 |
| LRT factor statistics for $PM_{2.5}$ | | | |
| $PM_{2.5}$[$\mu g/m^3$] | 11.14 | 6.77 | 6.46 |
| $PM_{2.5}$[%] | 37.61 | 33.93 | 30.85 |
| $PM_{2.5}$ BS 25th [$\mu g/m^3$] | 9.17 | 4.00 | 5.66 |
| $PM_{2.5}$ BS 50th [$\mu g/m^3$] | 10.53 | 5.63 | 6.53 |
| $PM_{2.5}$ BS 75th [$\mu g/m^3$] | 11.48 | 6.95 | 8.47 |

were excluded due to anomalous data. Overall, the 3 final PMF models selected for the analysis highlighted good performance, meeting the acceptance criteria with an RM in the range of 80 to 120 and an $R^2$ coefficient greater than 0.8 (Noris and Duvall, 2014). The results show a good convergence, where the $Q/Q_{expected}$ ratio for the selected base models was close to 1, a crucial parameter for determining the correct number of factors. A summary of the models' statistics is given in Table 2.

In evaluating errors of the PMF, none of the 3 simulated scenarios showed significant rotational ambiguity, nor were there substantial random errors in the dataset after running the DISP and BS methods. Results for the DISP method showed no factor swaps for all dQmax values. In the BS analysis, outputs were considered stable, yet not all base factors were mapped to the boot factors. On average, the percentage of factors correctly mapped was 84%, which is in line with (Noris and Duvall, 2014), where a minimum of 80% mapped factors are suggested for robust results and to support the number of factors selected. The target BB-LRT, Dust-LRT, and Volcanic-LRT profiles for the models were mapped in 88%, 80%, and 98% of the runs, respectively. The $25^{th}$, $50^{th}$ and $75^{th}$ percentiles of the $PM_{2.5}$ contribution rates calculated for the target profiles in the BS runs are presented in Table 2. The variability of the contribution factor ($g$) represented by the BS runs is particularly important

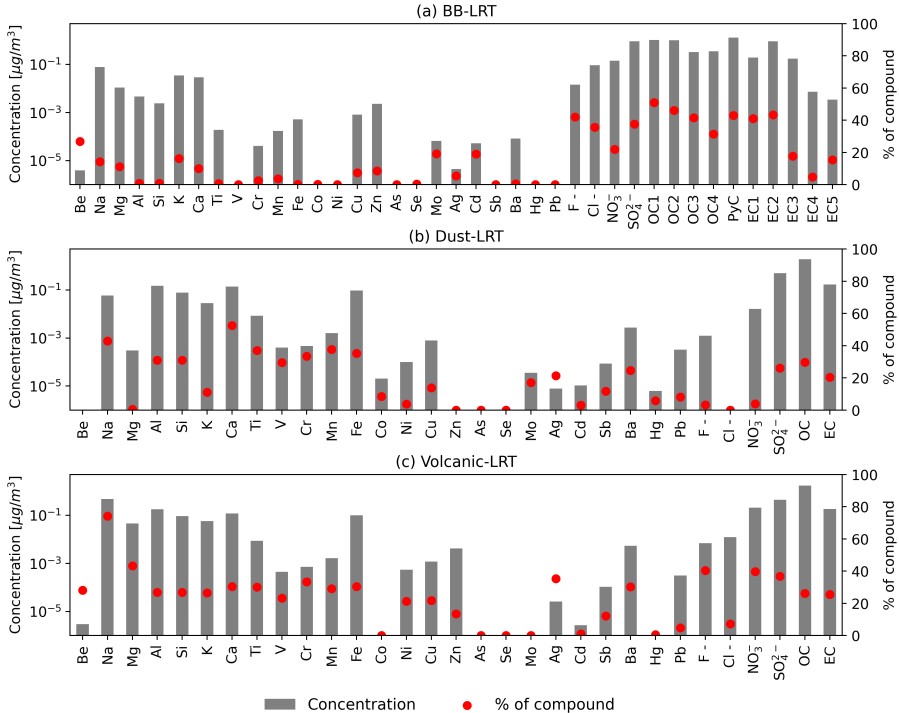

**Figure 8.** PMF output profiles for (a) BB-LRT, (b) Dust-LRT, and (c) volcanic-LRT events. The bars represent the average concentration of the compounds, while the red points denote the compound's average percentage contribution to the whole element concentrations. Refer to Supplementary Fig. S3 to see the contribution of all factors.

for the PMF model when using small dataset inputs (i.e., sample size < 100). Here, it will determine if the results from the model are stable or unstable (Feng et al., 2023).

Final base models were constrained to improve the correspondence between the chemical profiles found by the PMF and the profiles expected based on the identified emission sources. Specific constraints were defined in the different modeling scenarios to refine the factor profiles. The model's factors are presented in Fig. S3. For the BB-LRT model, three soft constraints were applied to the "coal boiler" factor to pull up the EC1, Se, and As concentrations. Conversely, the Ag, Se, and EC1 concentrations were pulled down in the "ceramic industry", "gasoline" and "diesel" factors. For the Dust-LRT model, four soft constraints were established. For "vehicular emissions", the concentrations of EC and Ni were maximally adjusted downward and upward, respectively. For "re-suspended materials," Ni was reduced to improve the fit; and for "biomass burning," OC was maximally increased. In the volcanic-LRT event scenario, two soft constraints were implemented. For volcanic-LRT, we pulled down the concentration of Cu and Mg, while Se was pulled up for "secondaries" and Cu was pulled down for "vehicular". For the models, the %dQ (i.e. the Q change because of the constraint) was < 1%.

The results from the PMF for BB-LRT ascribed concentration to six emissions sources (or factors): BB (37.6%), coal boiler (20.1%), ceramic industry (16.7%), gasoline (12.8%), diesel (11.4%) and incineration (1.4%). The factor of contribution for

BB-LRT in the days sampled ranged from 0.0 to 4.5 (maximum value identified on the 25 March 2020), which means that not all identified event days are backed up by the PMF (the factor of contribution from the PMF model runs are in Supplementary Fig. S4). Of the 31 samples identified, 27 days (equivalent to 87.0%) have positive contributions from BB-LRT events. The PMF profile for the BB factor, represented in Fig. 8a, is identified by the dominant contribution ($\geq$ 30%) of OC species, PyC and some anions. The identified contribution of this factor in OC, but especially in OC1 (50.9%) and OC2 (46.1%), agrees with observations made by Chow et al. (2004) for vegetation burning. Similarly, the high contribution of CyP (43.0%) supports the profile since BB produces $\sim$50% of global CyP emissions (Santín et al., 2016). Regarding anions, the high contribution to F$^-$ (42.1%) supports the identification of BB-LRT since this is a tracer of BB with a long lifetime (Jayarathne et al., 2014). The contribution to SO$_4^{2-}$ (37.7%) and NO$^{3-}$ (22.0%) suggests secondary pollutant formation from the BB emissions during the daytime and nighttime, respectively (Rastogi et al., 2014). Moreover, the influence of Cl$^-$ concentrations (35.7%) may be suggestive of semiarid vegetation burning (Andreae et al., 1998). Other species such as K (16.2%) are indicators of BB emissions (Yu et al., 2018; Rastogi et al., 2014).

For the Dust-LRT event days, the model identified PM$_{2.5}$ contributions from dust (33.9%) and BB (27.8%), vehicular (12.4%), resuspended material (6.5%), incineration (7.7%), and ceramic (11.7%). Due to only total OC and EC being considered, splitting the vehicular emissions between diesel and gasoline is difficult. The contribution factor for dust varies from 0.0 to 4.1 (maximum identified on 27 February 2020). For this event, 81.2% (13/16 samples) of the identified days exhibited a positive contribution. The dust profile classification is supported by a high contribution (>31.0%) of Al, Ca, Fe, and Si ($\geq$ 0.1 $\mu g/m^3$), as illustrated in Fig. 8b. Similarly, Malaguti et al. (2015) highlights Fe, Al, Ca, Ti, and Mg in the fine Saharan dust aerosols. Ca and Ti are specially weighted for LRT from the Sahara desert since the other elements are more generic crustal aerosol tracers (Martinez-Verduzco et al., 2023; Nicolás et al., 2008). The dust contributes to 52.6% and 37.0% of Ca and Ti concentrations, respectively. The profile also diverges from the general crustal material in the contribution of secondary inorganic aerosol as SO$_4^{2-}$ probably from (NH$_4$)$_2$SO$_4$ (Varrica et al., 2019; Malaguti et al., 2015), with 33.9% of average concentration in the samples. Likewise, a particular mix between dust minerals and OC (29.7%) might also be a characteristic to identify the dust profile (Aymoz et al., 2004; Malaguti et al., 2015).

For the days of Volcanic-LRT events, the model identified 30.8% of the PM$_{2.5}$ profile from volcanic aerosol contributions. Other sources were BB (29.1%), vehicular emissions (15.7%), resuspended dust and ceramic industry (11.0%), secondary aerosol formation (9.1%), and incineration (4.3%). The volcanic factor had contributions varying from 0.0 to 3.9 (maximum identified on 6 February 2020). According to the model, the volcanic factor influences only 78.1% (25/32 samples) of identified days. As expected, the volcanoes profile in Fig. 8c, shows a high contribution of SO$_4^{2-}$ (36.9%). However, the days affected by this LRT event did not exhibit significantly higher concentrations than the surrounding days. The contribution of the LRT event was higher for F$^-$ and NO$^{3-}$ than for SO$_4^{2-}$. F$^-$ from volcanic emissions frequently concentrates onto the surface of ash as CaF2, AlF3 or Na$_2$SiF$_6$ (Bia et al., 2020; Delmelle et al., 2021); while NO$^{3-}$ may relate to the oxidized tracer species HNO$_3$ (Martin et al., 2012). Minerals with relatively large concentrations in the identified profile include Na, Al, Ca, Fe, Si, K, and Mg in descending order presented concentration >0.04 $\mu g/m^3$ (contribution of >26.5%). The finding of Trejos et al. (2021) and Vanegas et al. (2021) supports the volcanic profile, with Si, Al, Fe, Ca, K, Mg, and Na as the main minerals for

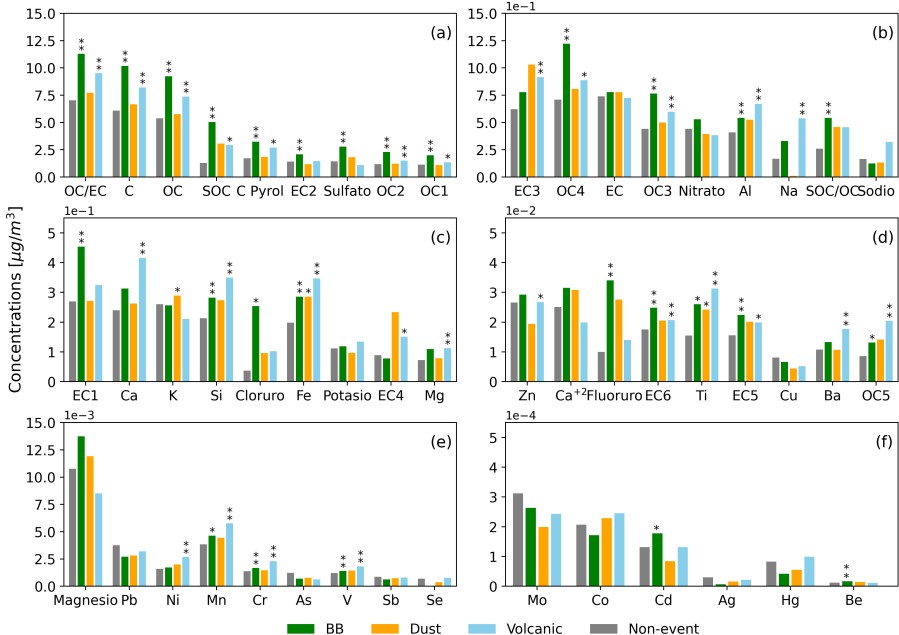

**Figure 9.** Campaign average concentration for non-event and positive-contribution days for BB-LRT, Dust-LRT, and Volcanic-LRT. The bars represent the concentration of the compounds along with the significance level compared with the days before and after the events, $\leq 0.1$ (*) or $\leq 0.05$ (**). The y-axes change for every figure to better identify concentrations. Supplementary Table S2 presents these values and the mean concentrations of the compounds for days around (i.e., before and after) each event.

the Nevado del Ruiz Volcano ashes. Although the composition might vary between volcanoes, the results for Sangay volcanic ashes in South America also backs up the profile with similar main elements (Moran-Zuloaga et al., 2023). From these, Fe, Al,

and Na are the most representative for the ash aerosols with a diameter $< 2.5 \mu m$ (Mason et al., 2021), while Ca, Na, Si, and K might be particularly persistent after LRT (Ruggieri et al., 2012), due to the atmospheric lifetime. Cu and Zn are other tracers observed here which have been identified before for the Colima Volcano in Mexico (Miranda et al., 2004).

   Figure 9 shows the average daily $PM_{2.5}$ concentrations of the campaign-measured data for each type of LRT event and when no event is identified. This only considers days with a positive contribution factor ($g_{i,k}$>0) from PMF results. Although

the non-event average concentration is presented, the statistical comparison is made between the LRT event period ($DtE_{-3}$ to $DtE_{+3}$) and the combined before and after event time series (i.e. the $DtE_{-15}$ to $DtE_{-4}$ period is combined with the $DtE_{+4}$ to $DtE_{+15}$ period) using the Mann Whitney U test. Unlike the PMF model, the comparison in Fig. 9 contains an analysis of the cations, the carbon matter species and the OC/EC and SOC/OC ratios for every type of event. Here, the major elements generally have a more significant increment in LRT events. Some elements supported the model's fingerprint (Fig. 8), e.g.,

OC, OC1, OC2, $SO_2^4$ for BB; Fe, Al, and Ti for dust; and Si, Al, Fe, Ca, Mg, and Na for volcanic aerosols (Fig. 9). The OC was significantly higher for BB, presenting a median OC/EC ratio of 11.3 that surpasses common urban combustion ratios

like from fossil fuel (∼4), combustion, and diesel exhausts (<1) (Pani et al., 2019). Although OC/EC is more commonly used to identify sources of urban combustion and BB, some studies have shown its potential for determining the influence of volcanic activity (Pongpiachan et al., 2019). This is supported by evidence of carbon enrichment resulting from volcanic

activity (Martinsson et al., 2009). This is also observed in this study, with consistently higher concentrations in all OC species and almost all EC species (EC3, EC4, EC5, EC6). For volcanic aerosols, OC/EC had a median ratio of 9.5. Likewise, the formation of SOC significantly increased for both events, with ratios of 5.0 and 2.9 for BB and volcanic aerosols, respectively. Heavy metals Cr (2.30 $ngm^{-3}$), Ni (2.67 $ngm^{-3}$), and Mn (5.76 $ngm^{-3}$) concentrations notably increased in event days for volcanic aerosols. Only the maximum daily concentration for Ni (25.05 $ngm^{-3}$) exceeded the air quality standards suggested

by the European Commission (2019) for the annual average concentration (20 $ngm^{-3}$). Increments for Mn (4.64 $ngm^{-3}$) and Cr (1.67 $ngm^{-3}$), in addition to Cd (0.18 $ngm^{-3}$), were observed for BB. The European Commission (2019) set an air quality standard of 5 $ngm^{-3}$ for the Cd annual average concentration.

The elevated concentrations of ions in the days of events (Fig. 9) also support the profiles (Fig. 8) and align with the literature. In addition to the ions observed in the PMF profile for the BB, the cations $K^+$ are representative of ions (Rastogi et al., 2014;

Moreno et al., 2023) that present significant increments for this type of event. Regarding the volcanic aerosol compositions (Fig. 9), the observed increment in $Na^+$ and $K^+$ also aligns with previous reports (Moreno et al., 2023; Mather et al., 2003; Roberts et al., 2018). On the other hand, although the PMF's fingerprint presented a high contribution of $SO2^{4-}$ and $F^-$, this was not enough for a significant rise in daily concentrations shown in Fig. 9.

## 4   Discussion

Our analysis focused on the LRT of aerosols from a range of sources (BB, dust and volcanic) to the AV. Long-term (i.e. several years) field measurements of $PM_{2.5}$ and its chemical composition helped to identify the impact of these events on local air quality. This long-term approach was important as it avoided restricting the study to short periods (i.e., sporadic singular large-scale high impact events). We used model/satellite-derived reanalysis data to identify regional-scale transboundary events and evaluated the effects on $PM_{2.5}$ using local aerosol field measurements.

Similar to other studies, thresholds were set for identifying LRT events (Ridley et al., 2012; Carn et al., 2008). However, their selection in this study was representative of local events rather than high-polluted sporadic events. The OM-AOD threshold for BB-LRT events (0.2) agrees with previous results (0.2-0.3) (Kaiser et al., 2012; Misra et al., 2020). The Du-AOD (0.02) threshold was lower (i.e., 0.05) than suggested by Ridley et al. (2012) and that used by Achilleos et al. (2020) to track outstanding episodes but is of a comparable order of magnitude. In contrast, the $TCSO_2$ threshold in this study for volcanic degassing

events differed from that discussed in the literature. The threshold used (0.87 $mg/m^2$) was more than 10 times than the used for another degassing volcano (17.15 $mg/m^2$) (Carn et al., 2008).

The lower $TCSO_2$ threshold derived in this study is likely linked to the CAMS product we used. While we use the CAMS reanalyses for OM-AOD, Dust-AOD, and $TCSO_2$ for consistency (i.e., tracers from the same model), the representation of the actual magnitude of $TCSO_2$ in the reanalysis product is lower than the operational product. The reanalysis uses an older

climatological (2005-2010) emissions dataset (CAMS-GLOB-VOLC) for volcanic degassing (Granier et al., 2019) and only assimilates satellite AOD. While satellite AOD is a useful product to help represent volcanic properties (e.g., volcanic plumes, $SO_2$ concentrations), the operational product also assimilates satellite $TCSO_2$ (CAMS, 2023), providing more constraint on the absolute magnitude of $TCSO_2$. However, regarding this study, the identification process is more important based on the data variability rather than the overall magnitude.

The identified BB-LRT reproduced the seasonality described in the literature (Mendez-Espinosa et al., 2019; Hernandez et al., 2019; Rodríguez-Gómez et al., 2022). Peak $PM_{2.5}$ concentrations occurred in February and March for BB-LRT events. In line with Mendez-Espinosa et al. (2019) and Henao et al. (2021), our findings show that the average conditions in this period are characterized by wind transport from the northeast of the AV, covering part of the Orinoco and Caribbean regions. BB aerosols from fires in the Orinoco can reach the AV through two passes in the Andes mountain range. The limited transport
throughout this complex topography potentially promotes the accumulation of aerosols, increasing their concentration in the AV (Ballesteros González, 2021). Furthermore, the back-trajectories highlight the the Middle Magdalena Valley, which suggests high BB emissions based on the reported OM-AOD hotspots. This area has been previously considered critical for open fires in Colombia (Bolaño-Díaz et al., 2022; Ballesteros González, 2021). According to Ballesteros González (2021), fires from this region are more likely to affect the Andean cities than fires from the Orinoco due to the mountain terrain acting as a natural
barrier to LRT.

The meteorology characterization for BB-LRT showed conditions favourable in enhancing LRT of aerosols to the AV. Here, dry conditions (i.e. low precipitation and humidity) in NSA promotes both the occurrence of fires and the LRT of pollutants. In contrast, despite the high emissions of Amazon fire smoke in the southern hemisphere, these do not considerably increase aerosols in the city due to the scavenging of particles during their transport (Hamburger et al., 2013). However, the BB-LRT
around August might be related to fires from that region (SIATA, 2021; Hamburger et al., 2013).

The estimated contribution to $PM_{2.5}$ concentrations during the BB-LRT showed the highest magnitude for the studied events, with 11.14 $\mu g/m^3$ (37.6%), supported by the comparison of $PM_{2.5}$ concentration. According to Ballesteros-González et al. (2020), the contribution from BB was calculated to be 4.7 $\mu g/m^3$ for Medellín in February 2010 and 2018. This result is consistent with an average (maximum) estimation of 6.0 (12.5) $\mu g/m^3$ for samples taken in the same month in 2020 from our
campaign, which is also supported by the PMF results. In the case of March 2020, at the beginning of the COVID-19 pandemic, the PMF estimated a maximum contribution of 50.0 $\mu g/m^3$ out of 60.9 $\mu g/m^3$. This particular event was characterized by a sudden reduction in local emissions but a sharp rise in $PM_{2.5}$ concentrations due to LRT from BB (Henao et al., 2021; Mendez-Espinosa et al., 2020).

Like BB-LRT, the monthly frequency of Dust-LRT events coincides with the annual cycle described by other studies. South
America experiences a dust reception from the Sahara desert mainly in two seasons, from March to May and June to August (Prospero et al., 2020). The seasonality of dust events partially coincides with the months of BB-LRT events, causing substantial overlapping between the identified days for these two types of events. Consequently, the identified weak impact on $PM_{2.5}$ by dust events controlled by Andes Mountain (Prospero et al., 2020) might be overshadowed by the most critical effect of BB-LRT. The only widely recognized event recorded for the study period occurred on June 24 and 25, 2020 (Mendez-Espinosa

et al., 2020; SIATA, 2021). According to the PMF results, this exhibited an almost 2 times higher contribution to $PM_{2.5}$ than the average event considered in this study. However, the PMF identified equivalent contributions in February 2020. The overlapping between these LRT event types, and the small database size for the Dust-LRT model, might result in less robust information from the PMF factor related to Saharan dust. However, the dust profile aligns with the literature; the organic carbon contribution likely contained contributions from combustion sources.

In contrast to BB-LRT and Dust-LRT events, the impact of Volcanic-LRT is expected to be primarily determined by the direction of the wind rather than an actual seasonality in the emissions. Volcanic-LRT events occurred more frequently from July to September, when low to mid-level winds flow from the Nevado del Ruíz volcano region in the southeast, facilitating the transport of emitted aerosols to the AV. According to the $SO_2$ V2 catalog, available on https://so2.gsfc.nasa.gov/measures.html (Fioletov et al., 2023), the AV is within the influence of the Nevado del Ruíz volcano. Indeed, this study identified a significant increment in $PM_{2.5}$ concentrations. According to the PMF results, on average, the contribution for $PM_{2.5}$ was 6.46 $\mu g m^{-3}$. Likewise, Casallas et al. (2024) linked the increase in $PM_{2.5}$ concentration during the JJA season with volcano activity in Cali, a relatively equidistant city located south of the volcano.

Although the Volcanic-LRT profile is predominantly made up by $SO_4^{2-}$, the total concentration of the elements do not show a significant increment compared with surrounding days (i.e. non-event days). This lack of significance in $SO_4^{2-}$ has been observed in other cities where urban sources of $SO_2$ dominate (Miyakawa et al., 2007). However, some alternative hypotheses relate to the magmatic gas state, suggesting that lower temperatures before emission might promote low sulfate oxidation and moderate $SO_4^{2-}$ formation (Mather et al., 2003) since $SO_4^{2-}$ precursors are formed at higher temperatures frequently close to the volcanic vent (Roberts et al., 2018). For instance, as observed in the Volcanic-LRT profile, the Cl-poor plumes might reduce sulfate oxidation and forward formation of $SO_4^{2-}$ (Mather et al., 2003).

The results have several implications for future research in the region. Modeling studies in the AV have only considered local emissions as inputs (e.g., Henao et al., 2020; Hernández et al., 2022). Therefore, it has been highlighted that external pollution sources need to be included as additional inputs into chemical transport models to have a more complete representation of air quality in the region. Further, this study suggests conducting shorter chemical campaigns and human health impact studies for target elements and sources, especially in the valley and upwind. For instance, special attention to the carbonaceous matter, $F^-$ and Cd emitted from BB in open fires is needed for human and ecosystem health studies in Colombia (Tuomisto et al., 2008; Jayarathne et al., 2014), since this source is particularly important for the region. The infrequent contribution of Saharan dust is still relevant, and there is particular interest in northern Colombia, where the most intense events significantly increases the mortality relative risk (Arregocés et al., 2023). On the other hand, Colombia has 12 active volcanoes, highlighting the importance of degassing activity in the country. Monitoring emissions and human health exposure is a primary need for municipalities, especially those close to active volcanoes such as Nevado del Ruíz. Both BB and volcanic events contribute to an increase in heavy metal concentrations. This can adversely impact human health when metal accumulation occurs in the body, leading to diseases such as thyroid cancer (Malandrino et al., 2020; Vigneri et al., 2017). The entry of heavy metals into the body via multiple routes (e.g. inhalation, ingestion and dermal absorption) increases the risk of exposure, which is more pronounced nearer the source (e.g., contamination of the air, water and crops). The increment in Mn and Cr might enhance

$PM_{2.5}$ toxicity in lung epithelial cells (Yuan et al., 2019). A deep assessment of the Cr(VI) and Cr(III) needs to be done to evaluate the correct toxicity status of Cr.

     Finally, this study highlights the need for cooperation among local, national, and international entities to manage complex aerosol sources/emissions and human exposure.

## 5    Conclusions

This study evaluates the impact of BB-LRT, Dust-LRT, and Volcanic-LRT aerosols on the $PM_{2.5}$ concentrations and its chemical composition in the AV, a densely populated and mountainous region in the tropical Andes. Events of the LRT of aerosols and their precursors (e.g. volcanic $SO_2$) were categorised when substantial enhancements (i.e. non-local) in the relevant tracers above background levels from the CAM reanalyses were identified over the campaign site in the AV. Meteorological data from ERA5 and GPM, along with a back-trajectories analysis, were used to link the aerosol events in the AV with the sources and to

characterize the meteorological patterns that determine the LRT of aerosols. The impact of these regional events on the local $PM_{2.5}$ was finally evaluated using field campaign measurements of local $PM_{2.5}$ and its chemical composition in conjunction with PMF modeling. This field campaign represents a unique long-term $PM_{2.5}$ composition dataset from April 2019 to October 2020.

     The methodology allowed for identifying periods influenced by the LRT of air pollutants, separating the days before, during,

and after the LRT events. During LRT event days, the back-trajectories identified air masses arriving from regions previously reported as critical sources of aerosols in the region. This enables the connection between aerosol events in the AV and the sources. The sources were identified as regional fires for BB-LRT, Saharan dust for Dust-LRT, and emissions from the Nevado del Ruíz volcano for volcanic-LRT. The different types of events were associated with substantial seasonal variability (i.e. pronounced season cylce). BB-LRT events occurred mainly in February, March, and April, with a lower occurrence during

August. Dust-LRT events showed a higher incidence from April to August. Volcanic-LRT showed a high occurrence from June to September and January to February. No events were identified during October, November and December.

     The weather patterns for BB-LRT events exhibited anomalous lower-tropospheric northeasterly winds, favoring regional transport of aerosols. Additionally, drier conditions in northern Colombia and Venezuela promoted the occurrence and spread of fires in these regions. For Dust-LRT, more intense winds from the Caribbean in the mid-troposphere favoured transport to

the AV when the aerosol loading was high. Moreover, during Volcanic-LRT events, anomalous southeasterly winds throughout the lower- and mid-troposphere were present, along with lower precipitation in the southeast of the AV, where the Nevado del Ruíz volcano is located, enabling the LRT.

     When evaluating the impact of LRT events on ground-level $PM_{2.5}$, BB-LRT events resulted in more significant increases in concentrations. The relative increment on $PM_{2.5}$ from Dust-LRT and Volcanic-LRT events was smaller but still prominent.

Similarly, underpinned by literature-supported profiles, the PMF results highlighted the importance of each LRT event type on influencing $PM_{2.5}$ chemical composition. OC species (e.g. OC1 and OC2), anions (e.g. $F^-$) and secondary pollutants (e.g. $SO_4^{2-}$ and $NO_3^-$) were the dominant components in the BB-LRT profile. On the other hand, general crustal minerals,

together with Ca and Ti, defined the Sahara-influenced Dust-LRT profile. Regarding Volcanic-LRT, $SO_4^{2-}$ and characteristic minerals (e.g. Na, Al, Ca, Fe, Si, K, and Mg) were substantially present in the corresponding profile. Further comparison of

the observations, on days which had a positive contribution factor, supported the PMF results although identified no substantial $SO_4^{2-}$ rise during volcanic degassing events. This aligned with the observed low $Cl^-$ contribution, supporting the reduced S oxidation hypothesis and forward production of $SO_4^{2-}$. Carbon and ion identification increments were mainly observed for BB-LRT and Volcanic-LRT. Lastly, the profiles associated with significant increments in the heavy metals Cr, Mn, Cd, and Ni highlight the potential risk to local air quality from LRT and are a cause for concern for local and national public health

authorities. Future studies using chemical models, larger event datasets, and more detailed measurement and characterization techniques may offer further explanations for this increase in observed heavy metal tracers.

The findings have several implications for future research. For instance, modeling studies in the region have only considered local emissions as inputs, and external pollution sources, especially volcanic degassing, need to be included as inputs in chemical transport modeling work/studies.

*Code availability.* A Python code for calculating the back-trajectories is available at https://github.com/MariaPa96/Basic_Trajectories_python.

*Data availability.* The CAMS reanalysis (Inness et al., 2019) data were downloaded from the Atmosphere Data Store at https://ads.atmosphere.copernicus.eu/cdsapp#!/dataset/cams-global-reanalysis-eac4?tab=overview. Meteorological data of the ERA5 reanalysis were downloaded from the Climate Data Store at https://cds.climate.copernicus.eu/cdsapp#!/dataset/reanalysis-era5-pressure-levels and GPM precipitation

data was retrieved from the NASA Earthdata at https://disc.gsfc.nasa.gov/datasets/GPM_3IMERGHH_07/summary. The hourly $PM_{2.5}$ concentration were download from the official network platform at https://siata.gov.co/siata_nuevo/. The datasets of wind for the trajectories calculation were downloaded from NCEP-NCAR Reanalysis 1 (https://psl.noaa.gov). The $PM_{2.5}$ chemical campaign datasets are available upon request to MGM (mgomez@elpoli.edu.co).

*Author contributions.* MPVG: Conceptualization, Methodology, Software, Formal analysis, Data Curation, Writing–Original Draft, Visu-

alization. KSH: Conceptualization, Methodology, Software, Writing-Original Draft, Writing - Review & Editing, Visualization. JAV: Data curation, Formal analysis, Software. RJP: Writing-review & editing. MGM: Conceptualization, Data Curation, Formal analysis, Methodology, Resources. AMR: Conceptualization, Methodology, Writing - Review & Editing, Supervision, Project administration.

*Competing interests.* The authors declare that they have no conflict of interest

*Acknowledgements.* For funding the PM$_{2.5}$ chemical composition sampling campaign, the authors would like to thank the project ONU-ARCAL (code: RLA7023) from the United Nations (UN) and the International Atomic Energy Agency (IAEA), the Área Metropolitana del Valle de Aburrá (contracts code: 734, 787 y 671), Ecopetrol (contract code: 3017481) and the Colombia Ministry of Science Technology and Innovation (BPIN:2020000100410). We also thank the Colombia Ministry of Science Technology and Innovation for funding the research work of Maria P. Velásquez-García and Miriam Gómez-Marím (BPIN:2020000100410). Additionally, we thank SIATA for providing air quality data and being a cradle of scientists in the territory.

Thanks to the research project "Implementación de un sistema de investigación y pronóstico meteorológico de corto plazo con el modelo WRF, para apoyo a sistemas de comando y control de la Fuerza Aérea Colombiana" (code 1115-852-70955) with funds of the "Patrimonio Autónomo Fondo Nacional de Financiamiento para la Ciencia, la Tecnología y la Innovación, Fondo Francisco José de Caldas" by the Colombian Ministry of Science, Technology and Innovation (MINCIENCIAS).

This work was also funded by the UK Natural Environment Research Council (NERC), which provided funding for the National Centre for Earth Observation (NCEO, award reference NE/R016518/1) and the NERC Panorama Doctoral Training Programme (DTP, award reference NE/S007458/1).

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
