# Peer review of "Assessing the influence of long-range transport of aerosols on the $PM_{2.5}$ chemical composition and concentration in the Aburrá Valley"

_EGUsphere, 2024_

## Referee Comment (RC1)

Review of Manuscript Entitled: **Assessing the influence of long-range transport of aerosols on the PM2.5 chemical composition and concentration in the Aburrá Valley**

**General comments:**

The manuscript presents the results of utilizing multiple tools, from in-situ PM2.5 chemical speciation data, source-receptor models, and back trajectory analysis, to estimate the monthly contribution of LRT to PM2.5 concentration in the Aburrá Valley, in Colombia. Furthermore, a careful characterization of the prevailing meteorological conditions during LRT events from Biomass Burning, Dust, and Volcanic degassing where also shown in the manuscript.

The manuscript is well written (some minor comments below) and advances the current understanding of the contribution of regional an global sources of PM2.5 in the Northern South American region. Particularly, the study points to the relevance of volcanic degassing for the region, an often overlooked source of aerosols, which is shown by the authors to be significant during LRT events.

The figures shown in the manuscript are of great quality.

As an overall recommendation, I would invite the authors to focus less on the importance of their findings for the AV, and rather focus on discussing the broader implications of their work for the region. That could be achieved with relatively ease but would require re-writing some specific parts of the document.

I recommend the manuscript to be published after addressing these minor comments.

**Specific comments:**

L10. "During these LRT events, the BB fraction of PM2.5 dominates by frequency and amount, averaging 11.14 µg/m3 (38%). On average, dust and volcanic degassing contribute 6.77 µg/m3 (34%) and 6.46 µg/m3 (30%) of the concentrations." This phrase might be confusing, specially the second part. What the authors really mean is that averaging over LRT events, dust and volcanic degassing contribute 34% and 30% of PM2.5? Is that the total? Something should be said in the abstract to at least provide the reader with an idea of the observed frequency of LRT events, or its typical duration (so those other numbers could be better contextualized).

L12. "Of the three, dust events showed fewer affected days." I would consider rewriting.

L63. "In Colombia, the Aburrá Valley (AV) has made substantial progress in monitoring and identifying agents of the state of air quality in the territory, managing to report significant affectation driven by external sources (SIATA, 2021)." Consider re-writing or removing altogether.

L64 "In the territory, as on the national scale" To which territory do the authors refer to? Please, consider removing or rewriting.

L65. There are at least 2 relevant studies in the region that could possibly enrich the discussion in the introduction:

- [https://doi.org/10.1016/j.atmosenv.2019.01.051](https://doi.org/10.1016/j.atmosenv.2019.01.051) which demonstrate the high concentration of PM2.5 and ozone in the Orinoco river basin during high BB seasons.
- [https://doi.org/10.5194/acp-20-7459-2020](https://doi.org/10.5194/acp-20-7459-2020) which shows the correlation of BB tracers with regional biomass burning activity.

L71. "To the AV, obtaining the …" Consider removing "To the AV".

L90. It would be useful to qualify this statement with data. For example, how many daily exceedances were observed in a given year? Or what is the annual mean PM2.5 concentration in the AV?

L107. "However, the sampling in this later period was typically between 3 to 14 days. Therefore, while the temporal sampling resolution did decrease with time, we still have periods of intense sampling and measurements across the majority of the period." Please rephrase as it is confusing.

L123. "Official campaign concentrations of PM2.5 were measured by a Low Volume PM2.5 ambient air sampler". Could the authors clarify this statement? What are "official" concentrations? How do the Low-vol concentrations differ from the High-volume sampler derived concentrations? Where the latter concentrations not determined at all? Please clarify and correct the manuscript accordingly.

L125. "In addition to the carbonaceous matter, species measured included secondary organic carbon". Please re-write for clarity. SOC was not measured, but it was inferred from the measurements.

L150. "mean absolute percentage error of 21.5%". Is this 21.5 percent overestimation relative to the MED-BEME station? Or 21% underestimation?

L187. "Here, if less than four days with values greater than the specified threshold were detected, then they were classed as outliers and removed (i.e., we are focusing on LRT events, which we define as lasting more than half a week". This is a key point in the manuscript and one that should be subject to a more specific description. Why focus on 4-day events? Dust events from LRT can impact a given location for a single day but contribute over 90% of PM2.5 to that given location on that day. If the decision is due to the sparsity of PM2.5 samples, then it should be clearly stated.

Figure 4. Caption and legend could be improved. No mention is made of the PM2.5 variable there. Is it monthly PM2.5 for the site? Or is ir PM2.5 attributable to LRT events? Similarly, the "All events" bar, which is black, it is not clear if there were any LRT events in which the three sources were impacting the site simultaneously.

L315. Seasonality?

L315. "some non-event days in the different months occur" ?? Please, consider re-writing for clarity.

L353. "On the other hand, the concentration of PM2.5 right after Volcanic-LRT significantly decreases" …. This assertion is hard to see from Figure 7c.

L442. "The lower TCSO2 threshold derived in this study is likely linked to the CAMS product we used". It is also possible that using SO2 observations (if available from the monitoring network) for the Volcanic-LRT events could help.

---

## Referee Comment (RC2)

**Assessing the influence of long-range transport of aerosols on the PM2.5 chemical composition and concentration in the Aburrá Valley**

In general, the manuscript shows the results of the chemical characterization of PM$_{2.5}$ by different techniques in addition to the identification of emission sources by using receptor models and back trajectory in the Aburrá Valley, in Colombia.

Specific chemical characterization during biomass burning and volcanic degassing processes were included in addition to dust events.

In general, the manuscript should be revised for minor errors.

The Introduction should provide context for the work in reference to past efforts to characterized fine particles and identify their sources in the Aburrá Valley.

I found quite confusing the way that the authors explain the modeled dates in Table 1, as they mentioned that is the samples number, but if it is the case, having 30 days is not enough for reliable PMF results. In addition, I do not understand why the eliminate the carbon fractions and trace elements and running the model just with anions which I do not thing is enough for the biomass burning and dust events.

Please give more details for the validation of the PMF results, specifically, some sensitivity analysis is needed to be better asses the robustness of the results.

The number of samples used for each site in each year should be presented. It is unclear if the number of samples is sufficient to draw the conclusions about trends across the sites and across years. If the meteorology is different for a majority of the samples in different years and there are different sources, then the comparison across years needs to be qualified.

**Specific comments:**

L12. Are the three dust events enough for the conclusions of the work. Consider re-writing.

L 90. A table with the days exceeding the PM2.5 should be included for each year and clarify if you are using the average the authors mean.

Authors should clarify what do they mean by official concentrations?

L 125: Secondary organic carbon can not be measured, as author stated, so this should be clarified.

Are the results statistically significant?

Figure 8 show the profiles identified, but they are not clear as the one identified as a Dust and volcanic have similar composition, how can the authors make sure of the name of the profile. More details should be given.

What is the purpose of the results in Figure 9?

I recommend the manuscript to be published after addressing these minor comments.

---

## Author Comment (AC1)

**Authors' Responses to Referee #2 Comments**

We gratefully acknowledge Referee #2 for their valuable and constructive comments, which helped improve the clarity and impact of this manuscript. We have reproduced their comments below in black text and we have number-listed Referee #2's comments for clarification when addressing comments relevant to both referees. Our responses are in blue text and any additions to the manuscript are in red text. Our reference to line numbers is based on the initially submitted manuscript.

1. L10. "During these LRT events, the BB fraction of PM2.5 dominates by frequency and amount, averaging 11.14 µg/m3 (38%). On average, dust and volcanic degassing contribute 6.77 µg/m3 (34%) and 6.46 µg/m3 (30%) of the concentrations." This phrase might be confusing, specially the second part. What the authors really mean is that averaging over LRT events, dust and volcanic degassing contribute 34% and 30% of PM2.5? Is that the total? Something should be said in the abstract to at least provide the reader with an idea of the observed frequency of LRT events, or its typical duration (so those other numbers could be better contextualized).

Thank you for your comment. Please see our response to Referee #2's Comment #2 below.

2. L12. "Of the three, dust events showed fewer affected days." I would consider rewriting.

To answer this comment and the previous one, we include the annual average frequency of the LRT events identified for the study period. This new information was added in L7 as:

"Annually, we found that on LRT of aerosols from BB, dust and volcanic degassing influence approximately 13, 8 and 13% of days, respectively."

3. L63. "In Colombia, the Aburrá Valley (AV) has made substantial progress in monitoring and identifying agents of the state of air quality in the territory, managing to report significant affectation driven by external sources (SIATA, 2021)." Consider re-writing or removing it altogether.

Following the comment, the sentence is removed and rewritten according to the idea of the paragraph. Now, this paragraph starts as:

"In the last few years, different studies in Colombia have made substantial contributions in monitoring and identifying impacts caused by open fire emissions on air quality (see e.g., Hernández et al., 2019; Mendez-Espinosa et al., 2019; Ballesteros-González et al., 2020; Rincón-Riveros et al, 2020; Henao et al., 2021). Nonetheless…"

4. L64 "In the territory, as on the national scale" To which territory do the authors refer to? Please, consider removing or rewriting.

We removed the text in L63 and changed it to consider Referee #2's Comment #5 below.

5. L65. There are at least 2 relevant studies in the region that could possibly enrich the discussion in the introduction:

- https://doi.org/10.1016/j.atmosenv.2019.01.051 which demonstrate the high concentration of PM2.5 and ozone in the Orinoco river basin during high BB seasons.
- https://doi.org/10.5194/acp-20-7459-2020 which shows the correlation of BB tracers with regional biomass burning activity.

We appreciate the provided references. As suggested, the papers have been included in different phrases in the introduction:

L20: "Long-range transport (LRT) of aerosols influences the chemical composition of air over hundreds to thousands of kilometers (Kaneyasu et al., 2014; Wang et al., 2015; **Rincón-Riveros et al., 2020**)"

L63: "In the last few years, different studies in Colombia have made substantial contributions in monitoring and identifying impacts caused by open fire emissions on air quality (see e.g., **Hernández et al., 2019;** Mendez-Espinosa et al., 2019; Ballesteros-González et al., 2020; **Rincón-Riveros et al, 2020;** Henao et al., 2021)."

L46: "Therefore, transboundary emissions from open fires in the Orinoco basin and the Caribbean are significant drivers of intra-annual periods of hazardous air quality for Colombian cities such as Bogotá, Medellín, Arauca, Yopal, Bucaramanga and Villavicencio (Mendez-Espinosa et al., 2019; **Hernandez et al., 2019; Rincón-Riveros et al., 2020;** Henao et al., 2021; Rodríguez-Gómez et al., 2022)…".

6. L71. "To the AV, obtaining the …" Consider removing "To the AV".

The recommendation was accepted, and L71 changed.

7. L90. It would be useful to qualify this statement with data. For example, how many daily exceedances were observed in a given year? Or what is the annual mean PM2.5 concentration in the AV?

We re-wrote the sentence in L91 to improve clarity and added a table in the supplemental material (Table S1) showing the average $PM_{2.5}$ concentration and daily $PM_{2.5}$ exceedances for each year of the study period (following Referee #1's Comment #2).

"… the average daily concentrations of $PM_{2.5}$ frequently exceed national and international standards (WHO, 2021, 15 ug/m$^3$) in the valley, with more than 60% of days exceeding this limit at most stations located in the urban areas of the AV (see Supplementary Table S1)."

Table added in supplementary material:

**Table S1**. PM$_{2.5}$ average concentration (ug/m$^3$) and percentage of exceedance of daily WHO [RP1] (15 μg/m$^3$) and national (37 μg/m$^3$) standards for urban stations from the Aburrá Valley's official air quality network during the study period.

| Official AQ Station | 2019 | | | 2020 | | | 2021 | | | 2022 | | |
|---|---|---|---|---|---|---|---|---|---|---|---|---|
| | Average | % WHO | % National | Average | % WHO | % National | Average | % WHO | % National | Average | % WHO | % National |
| CEN-TRAF | 28.6 | 99.2 | 14.2 | 25.5 | 87.2 | 10.9 | 25.0 | 97.0 | 1.9 | 26.7 | 98.6 | 5.5 |
| ENV-HOSP | 17.5 | 54.0 | 3.0 | 16.8 | 41.0 | 6.8 | 14.3 | 39.2 | 0.3 | 15.3 | 46.0 | 0.5 |
| EST-HOSP | 18.8 | 63.3 | 4.4 | 17.2 | 45.1 | 5.7 | 14.8 | 43.8 | 0.3 | 15.7 | 50.4 | 0.5 |
| ITA-CJUS | 23.4 | 91.5 | 6.8 | 22.7 | 79.0 | 10.7 | 20.8 | 87.7 | 1.4 | 23.7 | 90.7 | 4.9 |
| ITA-CONC | 18.5 | 66.3 | 2.7 | 19.6 | 64.2 | 6.8 | 16.6 | 61.4 | 0.3 | 16.7 | 47.1 | 0.3 |
| MED-ALTA | 22.7 | 92.6 | 5.2 | 19.9 | 69.7 | 6.0 | 18.6 | 79.7 | 0.5 | 19.4 | 81.6 | 1.1 |
| MED-ARAN | 22.2 | 86.0 | 7.4 | 20.1 | 62.8 | 9.6 | 17.8 | 68.2 | 0.5 | 19.3 | 77.8 | 0.5 |
| MED-BEME | 21.7 | 83.0 | 4.7 | 19.5 | 64.2 | 7.1 | 19.9 | 78.1 | 0.5 | 20.2 | 82.5 | 0.5 |
| MED-LAYE | 19.2 | 71.5 | 2.2 | 17.7 | 50.3 | 5.5 | 17.1 | 65.2 | 0.3 | 17.7 | 61.9 | 1.4 |
| MED-TESO | 17.2 | 54.0 | 3.3 | 16.8 | 42.3 | 6.8 | 14.9 | 45.8 | 0.3 | 15.7 | 50.1 | 1.1 |
| MED-VILL | 18.9 | 66.8 | 3.8 | 18.4 | 53.8 | 6.8 | 16.2 | 57.0 | 0.8 | 17.4 | 63.6 | 0.3 |
| SAB-RAME | 18.5 | 63.0 | 3.3 | 17.8 | 50.8 | 6.8 | 15.6 | 50.4 | 0.3 | 17.4 | 63.8 | 0.8 |

Note: this table includes the official air quality stations [RP2] in the most urbanized zone of the Aburrá Valley. The MED-BEME site corresponds to the location of the chemical sampling campaign (shadowed row).

8. L107. "However, the sampling in this later period was typically between 3 to 14 days. Therefore, while the temporal sampling resolution did decrease with time, we still have periods of intense sampling and measurements across the majority of the period." Please rephrase as it is confusing.

Following the suggestion, we rephrase the paragraph to provide clarity and additional information about the sampling frequency. The changes were implemented in L104

"While April 2019 to July 2020 represented an intense sampling campaign with samples every three days, the frequency of the surface site observations became less intense after July 23, 2020 (i.e. up to a maximum of 2 weeks during periods of routine sampling). However, there were two extended gaps in the campaign from November 2020 to mid-March 2021 and mid-September 2021 to March 2022. Despite the decrease in sampling frequency, the measurements still provide sufficient temporal coverage to get robust seasonal and annual information on aerosol concentration level and composition in this study."

9. L123. "Official campaign concentrations of PM2.5 were measured by a Low Volume PM2.5 ambient air sampler". Could the authors clarify this statement? What are "official" concentrations? How do the Low-vol concentrations differ from the High-volume sampler derived concentrations? Where the latter concentrations not determined at all? Please clarify and correct the manuscript accordingly.

Both measurements were part of the campaign but with different objectives. The Low-Vol equipment was used to measure the $PM_{2.5}$ concentrations during the campaign. This instrument met the Colombian national requirements (following the CFR 40 Appendix L to

Part 50 - US-EPA, 2017), so was classified as the "official" PM$_{2.5}$ sampler of the campaign. The Hi-Vol equipment was mainly used for the chemical characterisation of measured aerosol, which had the capacity to measure higher mass required for some of the chemical characterisation methods. The samples from this instrument were collected relative to the Australian/New Zealand Standard AS/NZS 3580.9.14:2013 Method 9.14 (Standards New Zealand, 2013).

The clarification was added to the paragraph in L124:

"The PM25 was additionally sampled by a Low-volume sampler (Reference: Wilbur TE-WILBUR - Tish). Since these measurements followed the reference method described by the 40 CFR Part 50 standards suggested by the US-EPA (2011) and adopted by Colombian regulations (MinAmbiente-Colombia, 2010), the calculated average 24-hour PM2.5 concentrations for the Low-volume are used for the positive matrix factorization model."

10. L125. "In addition to the carbonaceous matter, species measured included secondary organic carbon". Please re-write for clarity. SOC was not measured, but it was inferred from the measurements.

The sentence in L126 was changed to clarify that the SOC was calculated in the study.

"To complement the characterization of carbonaceous matter, the secondary organic carbon (SOC) was calculated using the elemental carbon trace methodology (Huntzicker et al., 1986)"

11. L150. "mean absolute percentage error of 21.5%". Is this 21.5 percent overestimation relative to the MED-BEME station? Or 21% underestimation?

Thank you for the observation; more information was added in L149 to interpret the equipment measurement differences better.

We found a good agreement between the campaign (low-vol sampler) instrument and the official MED-BEME station. For the study period, PM$_{2.5}$ concentrations from the automatic instrument had a minor overestimation against the reference method with a mean bias error of -0.76 µg/m3. The corresponding mean absolute error (MAE) was 21.5%. For PM$_{2.5}$ measurements, the low-volume as a reference method provides better precision and accuracy than the MED-BEME sensor (Tasić et al., 2012), which follows equivalent methods. Despite this, the official sensor provides continuous measurements that are used in this study for more robust comparisons. Regarding temporal variability, the Pearson correlation coefficient was 0.84, highlighting good consistency between them.

12. L187. "Here, if less than four days with values greater than the specified threshold were detected, then they were classed as outliers and removed (i.e., we are focusing on LRT events, which we define as lasting more than half a week". This is a key point in the manuscript and one that should be subject to a more specific description. Why focus on 4-day events? Dust events from LRT can impact a given location for a single day but contribute over 90% of PM2.5

to that given location on that day. If the decision is due to the sparsity of PM2.5 samples, then it should be clearly stated.

Thank you for the suggestion. The text of the paragraph was rewritten, and more details have been included to explain the selected filters more clearly from L187:

 "A 7-day rolling window was used to accurately identify prolonged and intense periods of LRT events. Within this window, at least 4 days had to have values above the respective thresholds to be classified as a LRT event. We subjectively chose 4 days of elevated values due to the sampling frequency of the campaign. Here, campaign temporal sampling was ≥ 3 days, so these criteria were required to get representative samples of the aerosol characterization for the PMF analysis."

**13.** Figure 4. Caption and legend could be improved. No mention is made of the PM2.5 variable there. Is it monthly PM2.5 for the site? Or is ir PM2.5 attributable to LRT events? Similarly, the "All events" bar, which is black, it is not clear if there were any LRT events in which the three sources were impacting the site simultaneously.

Thank you very much for noticing the problems in the caption. In accordance with the comments, we improved the caption to clarify:

"Figure 4. Monthly frequency of days with BB-LRT (green bars), Dust-LRT (wheat bars), and Volcanic-LRT (blue bars) events as identified from the CAMS reanalysis. Overlapped events are depicted in dark blue (BB and Volcanic), orange (BB and Dust), and red (Volcanic and Dust) bars since different LRT events can happen at the same time. White bars represent the frequency of days without LRT events, while the black line shows the monthly average $PM_{2.5}$ concentration ($\mu g/m^3$) for the MED-BEME station."

Regarding black bars, we removed this category from the figure's legend after verifying there is no overlapping among the three events.

**14.** L315. Seasonality?

Change accepted on L315. Please see our response to Referee #2's Comment #15 below.

**15.** L315. "some non-event days in the different months occur" ?? Please, consider re-writing for clarity.

The beginning of the paragraph was rewritten as follows (L315):

"Although LRT events display a marked seasonality, a significant percentage of days in each month have a negligible impact from LRT events (see white bars in Fig. 4), suggesting that intraseasonal variations are also relevant in explaining the occurrence of these events."

**16.** L353. "On the other hand, the concentration of PM2.5 right after Volcanic-LRT significantly decreases" …. This assertion is hard to see from Figure 7c.

The sentence was rewritten to explain more clearly what we wanted to highlight. The Mann-Whitney comparison supports a lower average concentration from 1 to 7 days after the peak of Volcanic degassing events. The following text will be added to L353:

"On the other hand, the concentration of PM2.5 immediately following the peak of Volcanic-LRT showed significantly lower levels (p-value ≤ 0.1), contrasting with the subsequent days ($Dte_8$ to $Dte_{15}$)."

**17.** L442. "The lower TCSO2 threshold derived in this study is likely linked to the CAMS product we used". It is also possible that using SO2 observations (if available from the monitoring network) for the Volcanic-LRT events could help.

The station does not have an $SO_2$ analyser, so we tried to use data from other stations in the city. However, the $SO_2$ record only had 38.53% valid data for the study period. Therefore, we decided not to include the pollutant.

**Added references**

AS/NZS: Methods for sampling and analysis of ambient air - Method 9.14: Determination of suspended particulate matter - PM2.5 high volume sampler with size selective inlet - Gravimetric method, https://www.standards.govt.nz/shop/asnzs-3580-9-142013/, 2013

Hernandez, A. J., Morales-Rincon, L. A., Wu, D., Mallia, D., Lin, J. C., & Jimenez, R. (2019). Transboundary transport of biomass burning aerosols and photochemical pollution in the Orinoco River Basin. *Atmospheric Environment*, 205, 1-8. https://doi.org/10.1016/j.atmosenv.2019.01.051

MinAmbiente-Colombia: MANUAL DE DISEÑO DE SISTEMAS DE VIGILANCIA DE LA CALIDAD DEL AIRE, https://www.minambiente.gov.co/wp-content/uploads/2021/06/Protocolo_Calidad_del_Aire_-_Manual_Diseno.pdf, 2010

Rincón-Riveros, J. M., Rincón-Caro, M. A., Sullivan, A. P., Mendez-Espinosa, J. F., Belalcazar, L. C., Quirama Aguilar, M., and Morales Betancourt, R.: Long-term brown carbon and smoke tracer observations in Bogotá, Colombia: association with medium-range transport of biomass burning plumes, *Atmos. Chem. Phys.*, 20, 7459–7472, https://doi.org/10.5194/acp-20-7459-2020, 2020.

Tasić, V., Jovašević-Stojanović, M., Vardoulakis, S., Milošević, N., Kovacević, R., and Petrovi̇́c, J.: Comparative assessment of a real-time particle monitor against the reference gravimetric method for PM10 and PM2.5 in indoor air, Atmospheric Environment, 54, 358–364, https://doi.org/https://doi.org/10.1016/j.atmosenv.2012.02.030, 2012

US-EPA: 40 CFR Appendix L to Part 50 - Reference Method for the Determination of Fine Particulate Matter as PM2.5 in the Atmosphere, https://www.govinfo.gov/app/details/CFR-2017-title40-vol2/CFR-2017-title40-vol2-part50-appL/context, 2011.

---

## Author Comment (AC2)

**Authors' Responses to Referee #1 Comments**

We thank Referee #1 for their valuable and constructive comments, which were helpful in improving the clarity and impact of this manuscript. Here, we have provided more information/discussion on our methods, particularly the PMF approach, to make it clearer to the reader the research we have done in this study. We have reproduced their comments below in black text and we have number-listed Referee #1's comments for clarification when addressing comments relevant to both referees. Our responses are in blue text and any additions to the manuscript are in red text. Our reference to line numbers is based on the initially submitted manuscript.

**Specific comments:**

**1.** L12. Are the three dust events enough for the conclusions of the work. Consider re-writing.

In this L12, we are talking about the three kinds of LRT events assessed, not about the number of events. Nonetheless, following this and Comment #2 from Referee #2, we clarified the statement, giving information about the annual frequency of the identified events for the study period.

**2.** L 90. A table with the days exceeding the PM2.5 should be included for each year and clarify if you are using the average the authors mean.

We re-wrote the sentence in L91 to improve clarity and added a table in the supplemental material (Table S1) showing the average $PM_{2.5}$ concentration and daily $PM_{2.5}$ exceedances for each year of the study period as suggested.

"... the average daily concentrations of $PM_{2.5}$ frequently exceed national and international standards (WHO, 2021, 15 ug/m$^3$) in the valley, with more than 60% of days exceeding this limit at most stations located in the urban areas of the AV (see Supplementary Table S1)."

Table added in supplementary material:

**Table S1.** $PM_{2.5}$ average concentration (ug/m$^3$) and percentage of exceedance of daily WHO [RP1] (15 µg/m$^3$) and national (37 µg/m$^3$) standards for urban stations from the Aburrá Valley's official air quality network during the study period.

| Official AQ Station | 2019 | | | 2020 | | | 2021 | | | 2022 | | |
|---|---|---|---|---|---|---|---|---|---|---|---|---|
| | Average | % WHO | % National | Average | % WHO | % National | Average | % WHO | % National | Average | % WHO | % National |
| CEN-TRAF | 28.6 | 99.2 | 14.2 | 25.5 | 87.2 | 10.9 | 25.0 | 97.0 | 1.9 | 26.7 | 98.6 | 5.5 |
| ENV-HOSP | 17.5 | 54.0 | 3.0 | 16.8 | 41.0 | 6.8 | 14.3 | 39.2 | 0.3 | 15.3 | 46.0 | 0.5 |
| EST-HOSP | 18.8 | 63.3 | 4.4 | 17.2 | 45.1 | 5.7 | 14.8 | 43.8 | 0.3 | 15.7 | 50.4 | 0.5 |
| ITA-CJUS | 23.4 | 91.5 | 6.8 | 22.7 | 79.0 | 10.7 | 20.8 | 87.7 | 1.4 | 23.7 | 90.7 | 4.9 |
| ITA-CONC | 18.5 | 66.3 | 2.7 | 19.6 | 64.2 | 6.8 | 16.6 | 61.4 | 0.3 | 16.7 | 47.1 | 0.3 |
| MED-ALTA | 22.7 | 92.6 | 5.2 | 19.9 | 69.7 | 6.0 | 18.6 | 79.7 | 0.5 | 19.4 | 81.6 | 1.1 |
| MED-ARAN | 22.2 | 86.0 | 7.4 | 20.1 | 62.8 | 9.6 | 17.8 | 68.2 | 0.5 | 19.3 | 77.8 | 0.5 |
| MED-BEME | 21.7 | 83.0 | 4.7 | 19.5 | 64.2 | 7.1 | 19.9 | 78.1 | 0.5 | 20.2 | 82.5 | 0.5 |
| MED-LAYE | 19.2 | 71.5 | 2.2 | 17.7 | 50.3 | 5.5 | 17.1 | 65.2 | 0.3 | 17.7 | 61.9 | 1.4 |
| MED-TESO | 17.2 | 54.0 | 3.3 | 16.8 | 42.3 | 6.8 | 14.9 | 45.8 | 0.3 | 15.7 | 50.1 | 1.1 |
| MED-VILL | 18.9 | 66.8 | 3.8 | 18.4 | 53.8 | 6.8 | 16.2 | 57.0 | 0.8 | 17.4 | 63.6 | 0.3 |
| SAB-RAME | 18.5 | 63.0 | 3.3 | 17.8 | 50.8 | 6.8 | 15.6 | 50.4 | 0.3 | 17.4 | 63.8 | 0.8 |

Note: this table includes the official air quality stations in the most urbanized zone of the Aburrá Valley. The MED-BEME site corresponds to the location of the chemical sampling campaign (shadowed row).

**3.** Authors should clarify what do they mean by official concentrations?

The PM$_{2.5}$ samples from the Low-Vol equipment were defined as the reference method following Colombian national requirements aligned with the CFR 40 Appendix L to Part 50 – (US-EPA, 2011), the reason why, for this study, this was agreed as the "official" PM$_{2.5}$ measurement for the campaign. But we agreed it was confusing, so clarify that in L124 as follows:

"The PM25 was additionally sampled by a Low-volume sampler (Reference: Wilbur TE-WILBUR - Tish). Since these measurements followed the reference method described by the 40 CFR Part 50 standards suggested by the US-EPA (2011) and adopted by Colombian regulations (MinAmbiente-Colombia, 2010), the calculated average 24-hour PM$_{2.5}$ concentrations for the Low-volume are used for the positive matrix factorization model."

**4.** L 125: Secondary organic carbon cannot be measured, as author stated, so this should be clarified.

The sentence in L126 was changed to clarify that the SOC was calculated in the study.

"To complement the characterization of carbonaceous matter, the secondary organic carbon (SOC) was calculated using the elemental carbon trace methodology (Huntzicker et al., 1986)"

**5.** Are the results statistically significant?

Following the body of the comment, we understand that this question targets the PMF results with concern about the size of the datasets. We recognize that the sample size is a critical

factor for the PMF results' reliability and we did previously consider the factors/metrics for the uncertainties for this method. As a result, we have added the text below to improve the manuscript. Two additional rows were also added to Table 2.

For Methodology section in L253

[revised manuscript text omitted]

In addition, this study includes different independent methodologies aiming to add robustness to the appreciated small dataset PMF assessment. For instance, a partial answer to one comment not included in the "specific comment", the delivery described as the monthly frequency used the daily average CAMS's products instead of the samples; the back trajectories and a meteorological assessment was made as complements. The PM$_{2.5}$ comparison in Figure 7 includes the daily average PM$_{2.5}$ concentration, and after presenting the PMF result, the comparison of all measured compounds was made in Figure 9.

**6.** Figure 8 shows the profiles identified, but they are not clear as the one identified as a Dust and volcanic have similar composition, how can the authors make sure of the name of the profile. More details should be given.

Targeting Referee #1's main comment, we wanted to clarify first that all measured elements were used for dust and volcanic aerosols. The campaign did not measure some key trace elements from April 2021, including K, Mg, and Na concentrations. Because these elements are trace elements for these two factors, the decision was to restrict the samples to the period where all elements were measured instead of removing them from the model, as is explained in L253. The carbonaceous species were removed, but the total OC and EC were still included in the models, considering the elements, not carbon, as the target compounds for these sources.

Given the diverse chemical structures of the sources and the mixing between them due to the different phenomena associated with atmospheric dispersion and dynamics, identifying physically significant profiles is based on groups of chemical components called pseudo components according to the relevant species or sources. That is why a mixture of sources is often a constraint. To accurately label the factors, it is crucial to utilize tracers supported by literature and thoroughly evaluate the sources at the site. The site sources assessment is backed by Gómez et al. (2021) (as L257 says) and the first part of the study for LRT events. Dust and volcanic aerosol share key tracers identified as crustal tracers, and those have marked contributions in the majority of compounds. In addition to Figure 8, Figure S3 is a good complement for appreciating what is described for the profile identification and then recognizing key differentiation compounds better. As described in L394, in addition to the crustal tracers, the contribution of the anions led by F-, NO3, and SO4, together with Na and K, are crucial trace compounds for the Volcanic-LRT factor. Meanwhile, for the Dust-LRT factor, Ca and Ti were key minerals (described from L384).

It is important to highlight that both factor profiles belong to independent models, so differentiating one from the other was not a direct necessity. Besides, other analyses supported both events, helping us back up the results.

An additional reference was added to back up the volcanic factor selection:

"Cu and Zn are other tracers observed here and identified before for Colima Volcano in the southeast of the ring of fire (Miranda et al., 2004)."

**7.** What is the purpose of the results in Figure 9?

Figure 9 has two main objectives. The first objective is similar to Figure 7 for $PM_{2.5}$, comparing the compound concentrations for days affected by a LRT event and the closest and most meteorologically similar days (i.e., days before and after the event), as described in L404. The second is present cation concentration and some index magnitudes (e.g., OC/EC) not included in the PMF, as described by L409. Since the campaign had a big discontinuity in the cations measurement, these were not used in the PMF for the profile characterization. This is part of the methodologies that, as was explained at the beginning of the document, aim to increase robustness in the analysis, in this case, for the selected profile.

We noticed that the connection between L409 and L424 and Figure 9 might not be evident, so we have rewritten those sections, including clearly referencing the figure.

"Unlike the PMF model, the comparisons in Fig. 9 contain analysis of the cations, the carbon matter species and the OC/EC and SOC/OC ratios for every type of event. Here, the major elements generally have a more significant increment in LRT events. Some elements supported the model's fingerprint (Figure 8), e.g., OC, OC1, OC2, SO4 2 for BB; Fe, Al, and Ti for dust; and Si, Al, Fe, Ca, Mg, and Na for volcanic aerosols (Figure 9). The OC was significantly higher for BB, presenting a median OC/EC ratio of 11.3 that surpasses common urban combustion ratios like from fossil fuel (~4), combustion, and Diesel exhausted (<1) (Pani et al., 2019). Although OC/EC is more commonly used to identify sources of urban combustion and BB, some studies have shown its potential for determining the influence of volcanic activity (Pongpiachan et al., 2019)."

"The elevated concentrations of ions in the days of events (see Fig. 9) also support the modeled profiles (Fig. 8) and align with the literature. In addition to the ions observed in the PMF profile for the BB, the cations K+ are representative ions (Rastogi et al., 2014; Moreno et al., 2023) that present significant increments for this type of event. Regarding the volcanic aerosol compositions (Fig. 9), the observed increment in Na+ and K+ also aligns with previous reports (Moreno et al., 2023; Mather et al., 2003; Roberts et al., 2018). On the other hand, although the PMF's fingerprint presented a high contribution of $SO_2^{4-}$ and $F^-$, this was not enough for a significant rise in daily concentrations shown in Fig. 9."

**Added references**

Feng, J., Song, N., & Li, Y. (2023). An in-depth investigation of the influence of sample size on PCA-MLR, PMF, and FA-NNC source apportionment results. *Environmental Geochemistry and Health*, *45*(8), 5841–5855. https://doi.org/10.1007/s10653-023-01598-5

Haghnazar, H., Johannesson, K. H., González-Pinzón, R., Pourakbar, M., Aghayani, E., Rajabi, A., & Hashemi, A. A. (2022). Groundwater geochemistry, quality, and pollution of the largest lake basin in the Middle East: Comparison of PMF and PCA-MLR receptor models and application of the source-oriented HHRA approach. *Chemosphere*, *288*, 132489. https://doi.org/10.1016/j.chemosphere.2021.132489

Manousakas, M., Papaefthymiou, H., Diapouli, E., Migliori, A., Karydas, A. G., Bogdanovic-Radovic, I., & Eleftheriadis, K. (2017). Assessment of PM2.5 sources and their corresponding level of uncertainty in a coastal urban area using EPA PMF 5.0 enhanced diagnostics. *Science of The Total Environment*, *574*, 155–164. https://doi.org/10.1016/j.scitotenv.2016.09.047

MinAmbiente-Colombia: MANUAL DE DISEÑO DE SISTEMAS DE VIGILANCIA DE LA CALIDAD DEL AIRE, https://www.minambiente.gov.co/wp-content/uploads/2021/06/Protocolo_Calidad_del_Aire_-_Manual_Diseno.pdf, 2010

Noris, G. and Duvall, R.: EPA Positive Matrix Factorization (PMF) 5.0 Fundamentals and User Guide, https://www.epa.gov/sites/production/files/2015-02/documents/pmf_5.0_user_guide.pdf, 2014

Via, M., Chen, G., Canonaco, F., Daellenbach, K. R., Chazeau, B., Chebaicheb, H., Jiang, J., Keernik, H., Lin, C., Marchand, N., Marin, C., O'Dowd, C., Ovadnevaite, J., Petit, J.-E., Pikridas, M., Riffault, V., Sciare, J., Slowik, J. G., Simon, L., … Cruz Minguillón, M. (2022). Rolling vs. seasonal PMF: real-world multi-site and synthetic dataset comparison. *Atmospheric Measurement Techniques*, *15*(18), 5479–5495. https://doi.org/10.5194/amt-15-5479-2022

Yu, W., Liu, R., Wang, J., Xu, F., & Shen, Z. (2015). Source apportionment of PAHs in surface sediments using positive matrix factorization combined with GIS for the estuarine area of the Yangtze River, China. *Chemosphere*, *134*, 263–271. https://doi.org/10.1016/j.chemosphere.2015.04.049

Zhang, Y., Sheesley, R. J., Bae, M.-S., & Schauer, J. J. (2009). Sensitivity of a molecular marker based positive matrix factorization model to the number of receptor observations. *Atmospheric Environment*, *43*(32), 4951–4958. https://doi.org/10.1016/j.atmosenv.2009.07.009

---

## Author Response (AR2)

**Authors' Responses to Editor's comments**

We appreciate the editor's minor and technical comments, as we understand that addressing these will elevate the quality and impact of our manuscript and its consistency with the journals' articles. Similar to the answer to reviewers' comments, here, our response is presented in blue color, and any highlighted change to the manuscript is in red.

**Minor comments:**

1. Title: Please Adjust it following the ACP guidelines ("Titles should be concise and consistent with the content and purpose of the article. For research articles, ACP prefers titles that highlight the scientific results/findings or implications of the study.")

After the discussion, we selected the following title, which we believe suits the ACP guideline and the manuscript description.

"Long-range transport of air pollutants increases hazardous components of PM2.5 in northern South America"

2. The grammar needs to be significantly improved. It seems that different people wrote different parts of the manuscript as the readability of the entire document is not homogenous. Please make sure to improve the readability of the entire document, ideally by a native English person. Below you can find some examples of sentences or paragraphs that need to be improved; however, the list is much longer.

The manuscript was edited to improve readability, as suggested. We also followed apart the technical comment in the second part of this answer.

3. Line 44: "Three primary kinds of LRT of aerosols have been identified in the region". How about marine aerosol?

In this part, we referred to the LRT of aerosols impacting air quality in the Colombian Andes. In this sense, we rewrote the sentence and specify for clarity:

"Particularly in the Colombian Andes, biomass burning, desert dust, and volcanic emissions have been identified as three main sources of aerosols that can impact air quality from distant regions."

4. Figure 2. I suggest using more contrasting colors.

We changed the lines' colors to contrast as suggested and thickened the lines to improve visibility.

5. Follow the ACP guidelines to call Tables and Figures in the main text.

Following the guidelines, we changed "Fig." on L319, L388, L473 and L321 for "Figure" as the figures calls began a sentence. On the contrary, we changed "Figure" to "Fig." on

L330 and L371 because they were in-text calls. In addition, Fig. 8 captions L436 and L330, "Figure S," were changed to "Fig. S" as in-text calls.

The abbreviation "Tab." was changed to "Table" on L312 and L323.

6. Figure 8. Maybe "fraction" is more appropriate than "proportion"? It would be nice to be consistent using the brackets in all figures containing multiple panels.

We decided to use "% of compound" to match the legend and not to give space for misunderstanding. Moreover, we used brackets for figures with multiple panels in both the main manuscript and the supplementary material.

**Technical comments:**

We express our gratitude to the editor for their valuable technical comments, which we have carefully considered. We agreed with the suggested changes or rephrased unclear sentences. The accepted changes are checked (crossed) in the following list.

-
-
- L9: What do the authors mean with "secondary aerosols trace"?
  correction: "secondary aerosol tracers"
- L16: "the Caribbean for dust". Is this correct? I think the original source are the deserts in northern Africa and not the Caribbean
  correction: "Our study identifies the Orinoco and Middle Magdalena Valley as sizeable sources of BB aerosols and the Nevado del Ruíz volcano for volcanic aerosols. Additionally, we found that African dust approached the Andean region via the Caribbean route."
-
-
-
-
-
-
-
-
-
- L83: The following sentence is unclear: "chemical composition through year-frequent high aerosol load events"
  The sentence was rewritten for clarity: "This study aims to analyze the impact of inter-annual LRT of biomass burning (BB-LRT), dust (Dust-LRT), and volcanic aerosols (Volcanic-LRT) on PM$_{2.5}$ concentrations and chemical composition in the Aburrá Valley"
- L85: "favorable conditions for transport". Transport of what?

The sentence was rewritten for clarity: "We characterize meteorologically favorable conditions for the LRT events and involve information from one of the largest registered PM$_{2.5}$ chemical characterization campaigns conducted in the region."

- L86: Replace "conducted in the territory" by "conducted in the region"
-
- L107: Add a reference after "PM10"

  As suggested, a reference was added: "These conditions modulate the intra-annual variability of PM2.5 and PM10 (Mendez-Espinosa et al., 2019)"
-
-
-
-
- L135: "mal/optical transmission (TOT)" is a technique not a sensor

  Correction: "Distinct analytical methodologies were applied to determine the concentration of minerals, carbonaceous matter, and ions in the filters. An Inductively Coupled Plasma Mass Spectrometry (ICP-MS) methodology was used for minerals, a thermo optical transmission (TOT) methodology for carbonaceous matter, and an ionic chromatography (IC) for both anion and cations."
-
- L194: "a back trajectories dataset was built from the chemical sampling point". What does it mean?

  The sentence was rewritten for clarity: "a back trajectories dataset was built to estimate pollutants arriving at the chemical sampling point in the AV"
- L204-203: The following sentence is unclear "Furthermore, its products relatively reasonably reproduce PM2.5"

  The sentence was rewritten for clarity: "Furthermore, the CAMS' products reasonably capture PM$_{2.5}$ tendencies and extreme events in the territory (Casallas et al., 2022)"
-
-
-
- L226: The following is unclear "during pollution LRT events to identify conditions that"

  The sentence was rewritten for clarity: "A regional analysis of meteorological fields was performed during the LRT events to identify atmospheric conditions that favor aerosol transport from the sources of interest."
-
- L248: The following is unclear "different LRT classed events"

  The sentence was rewritten for clarity: "... during each LRT event"
- L303: The following is unclear "final element comparison"

  The sentence was rewritten for clarity: "A final comparison of the characterized compounds was conducted, focusing on the days with a positive contribution occurred for the LRT events"
-
-

-
- L325: What do the authors mean with "identification threshold"

  correction: "identification" was removed
-
-
- L390: Replace "Besides" by "Moreover" or something similar
-
- L441-442: What do the authors mean with "head contribution"?

  "head contribution" was remove and the sentence was rewrote: "Regarding anions, the high contribution to F⁻ (42.1%) supports the identification of BB-LRT since this is a trace with a long lifetime (Jayarathne et al., 2014)"
-
- L462-463: The following is unclear "The contribution but not the concentration of SO2– was surpassed by F– and NO3–."

  The sentence was rewritten for clarity: "As expected, the volcanoes profile in Fig. 8 c, shows a high contribution of $SO_4^{2-}$ (36.9\%). However, the days affected by this LRT event did not exhibit significantly higher concentrations than the surrounding days"
- L491-492: What do the authors mean with "suggested for the annual average of the compound"?

  The sentence was rewritten for clarity: ". Only the maximum daily concentration for Ni (25.05ngm−3) exceeded the air quality standards suggested by the European Commission (2019) for the annual average concentration (20 ngm$^{-3}$)."
-
- L494: The following is unclear "for the Cd average."

  The sentence was rewritten for clarity: "The European Commission (2019) set an air quality standard of 5 ngm$^{-3}$ for the Cd annual average concentration."
-